# Gonadal sex patterns p21-induced cellular senescence in mouse and human glioblastoma

Lauren Broestl[1], Nicole M. Warrington[1], Lucia Grandison[1], Tamara Abou-Antoun[1], Olivia Tung[1], Saraswati Shenoy[2], Miranda M. Tallman[3,4], Gina Rhee[1], Wei Yang[5], Jasmin Sponagel [1], Lihua Yang[1], Najla Kfoury-Beaumont[1,6], Cameron M. Hill[1], Sulaiman A. Qanni[1], Diane D. Mao[7,8,9], Albert H. Kim [7,8,9], Sheila A. Stewart [9,10,11,12], Monica Venere [3], Jingqin Luo [13] & Joshua B. Rubin [1,8✉]

Males exhibit higher incidence and worse prognosis for the majority of cancers, including glioblastoma (GBM). Disparate survival may be related to sex-biased responses to treatment, including radiation. Using a mouse model of GBM, we show that female cells are more sensitive to radiation, and that senescence represents a major component of the radiation therapeutic response in both sexes. Correlation analyses revealed that the CDK inhibitor p21 and irradiation induced senescence were differentially regulated between male and female cells. Indeed, female cellular senescence was more sensitive to changes in p21 levels, a finding that was observed in wildtype and transformed murine astrocytes, as well as patient-derived GBM cell lines. Using a novel Four Core Genotypes model of GBM, we further show that sex differences in p21-induced senescence are patterned during early development by gonadal sex. These data provide a rationale for the further study of sex differences in radiation response and how senescence might be enhanced for radiation sensitization. The determination that p21 and gonadal sex are required for sex differences in radiation response will serve as a foundation for these future mechanistic studies.

[1] Department of Pediatrics, Washington University School of Medicine, St. Louis, MO, USA. [2] Brown School, Washington University in St. Louis, St. Louis, MO, USA. [3] Department of Radiation Oncology, James Cancer Hospital and Comprehensive Cancer Center, The Ohio State University Wexner School of Medicine, Columbus, OH, USA. [4] Biomedical Sciences Graduate Program, The Ohio State University, Columbus, OH, USA. [5] Department of Genetics, Washington University School of Medicine, St. Louis, MO, USA. [6] Department of Neurological Surgery, University of California San Diego, La Jolla, CA, USA. [7] Department of Neurosurgery, Washington University School of Medicine, St. Louis, MO, USA. [8] Department of Neuroscience, Washington University School of Medicine, St. Louis, MO, USA. [9] Siteman Cancer Center, Washington University School of Medicine, St. Louis, MO, USA. [10] Department of Cell Biology and Physiology, Washington University School of Medicine, St. Louis, MO, USA. [11] Department of Medicine, Washington University School of Medicine, St. Louis, MO, USA. [12] ICCE Institute, Washington University School of Medicine, St. Louis, MO, USA. [13] Department of Surgery, Washington University School of Medicine, St. Louis, MO, USA. ✉email: Rubin_j@wustl.edu

Sex differences are observed in the majority of diseases, including cancer. Across a wide range of ages and cultures, male sex is associated with higher incidence and a worse prognosis for most tumor types[1–3]. Glioblastoma (GBM), the most common primary malignant brain tumor, follows this same pattern—women are both less likely to develop GBM and have a significant survival advantage compared to men[4–6]. Sex differences in survival may result from differences in the response to standard of care therapy, which for GBM includes surgical resection, followed by treatment with radiation and chemotherapy. Previous research from our lab found that female GBM patients had a greater decline in tumor growth velocity after treatment with radiation and chemotherapy than male patients, and that when these measures were used to stratify patients, there was a significant association with survival in female but not male patients[7]. Furthermore, we identified unique molecular pathways associated with improved survival in male and female patients, and the expression of genes in these pathways correlated with the sensitivity to a range of chemotherapeutic agents[7]. While this study advanced our understanding of the relationship between chemotherapy and cellular responses in male and female GBM, it did not focus on radiation, the backbone of GBM treatment, which is currently applied to male and female patients equally.

In cancer treatment, the goal is to stop tumor cell proliferation. While one mechanism to achieve this is through triggering cell death/apoptosis, another possibility is to activate cellular senescence—a cell fate decision that results in irreversible cell cycle arrest[8]. Senescence is a known outcome of radiation treatment, and at least one study has reported that this is the dominant response in glioblastoma[9]. Senescence is often described as a double-edged sword, since senescent cells secrete a broad variety of factors with complex pro- and anti-tumorigenic effects[10–12]. However, the cell-intrinsic aspect of senescence, specifically the terminal cell cycle exit of cells harboring mutated DNA, serves a beneficial function in tumor prevention and treatment, and enhancing this response could be a strategy to improve treatment efficacy.

Senescence is primarily regulated by two central pathways: p53/p21$^{WAF1/Cip1}$ and p16$^{INK4A}$/Rb[8,10,11]. Importantly, we have previously identified sex differences in the regulation of p21, p16, and Rb in a GBM model, with female cells being more likely to upregulate these pathways in response to cellular stress[13,14]. Whether these differences influence the cellular response to radiation in males and females is currently unknown. In this study, we show that primary male and female human GBM lines have unique molecular pathways that contribute to radiation sensitivity. Using an in vitro mouse model of GBM, we find that females are more sensitive to radiation, and that senescence is a major component of the radiation response in both males and females. Using a correlation-based approach, we identify an association between p21 and irradiation-induced cellular senescence that differs in males and females. The same levels of p21 correlate with higher percentages of senescent cells in females than in males—a finding that was observed in both wild-type and transformed cells, as well as in patient-derived GBM cell lines. Finally, using a novel Four Core Genotypes (FCG) model of GBM, we investigate the biological mechanisms underlying this sex difference, and show that sex differences in p21-induced senescence are patterned by gonadal sex.

## Results

### Unique molecular pathways contribute to radiation response in male and female primary human GBM cell lines. Female GBM patients exhibit significantly longer median survival than

male patients[5,15]; concordantly, we previously determined that, compared to males, female GBM patients exhibited a greater initial therapeutic response to radiation and temozolomide therapy[7]. Moreover, the degree of initial response to chemoradiation was significantly correlated with survival in female, but not male, patients. To investigate the molecular features underlying this sex difference in treatment response, we applied a Joint and Individual Variance Explained (JIVE) algorithm to identify and filter out gene expression patterns that were shared between male and female patients. This revealed unique male and female gene signatures that correlated with sex differences in survival, as well as with sensitivity to a range of chemotherapeutic agents in primary human GBM cell lines. Since our long-term goal is to determine whether survival with GBM can be enhanced by adapting treatment to sex differences in biology, to further explore the clinical relevance of these gene signatures, we investigated whether they similarly correlated with sensitivity to radiation.

We performed irradiation dose-response curves with four male and five female primary human GBM lines (Supplementary Table 1; Supplementary Fig. 1), and calculated an individual IC$_{50}$ value for each line. As previously observed for other DNA damaging agents, there was no significant difference in the median IC$_{50}$ values for male and female lines (Fig. 1a).

Our previous work identified a set of differentially regulated genes associated with better survival in either male (Male Cluster 5—MC5) or female (Female Cluster 3—FC3) GBM patients (see Fig. 3a, b and Fig. 5c, d in ref. [7]). Pathway analysis of these gene sets identified enrichment of a cell cycle regulation pathway in males (MC5—17 genes) and an integrin signaling pathway in females (FC3—9 genes) (Supplementary Table 2). In both cases, downregulation of the majority of genes in the pathway was associated with improved survival[7]. We previously observed that low expression of the MC5 gene set was associated with low IC$_{50}$ for multiple chemotherapies in male cell lines, while low expression of the FC3 gene set was associated with low IC$_{50}$ for multiple chemotherapies in female cell lines—a finding consistent with the observation that downregulation of the MC5 and FC3 genes was associated with improved survival in male or female GBM patients, respectively[7]. To determine whether these molecular pathways may also be contributing to the male and female irradiation response, we calculated Spearman rank correlation coefficients between IC$_{50}$ values and the expression levels for MC5 and FC3 genes, measured using the Illumina HumanHT-12 v4 expression microarray. For the female cell lines, we saw a similar pattern to that observed with other DNA damaging agents (see Fig. 7b, c in ref. [7]). There was a mild positive correlation between FC3 gene expression and irradiation IC$_{50}$, indicating that low expression of these genes was associated with low IC$_{50}$, or better response to irradiation. There was also a negative correlation between MC5 gene expression and IC$_{50}$, indicating that high expression of the MC5 genes was associated with low IC$_{50}$ (i.e. better response to irradiation) in the female lines (Fig. 1b). In contrast, the male cell lines showed no significant correlation between MC5 gene expression and IC$_{50}$, while there was a significant negative correlation between FC3 gene expression and IC$_{50}$, indicating that high expression of the FC3 genes was associated with better response to irradiation in the male lines (Fig. 1b). Neither male nor female lines showed any significant correlation between IC$_{50}$ and the expression levels of randomly selected equally sized gene sets, serving as a negative control. These results suggest that the response to radiation therapy may be driven by different intracellular pathways in males and females, and that efforts to improve response to therapy may be advanced by identifying the mechanism(s) underlying this sex difference.

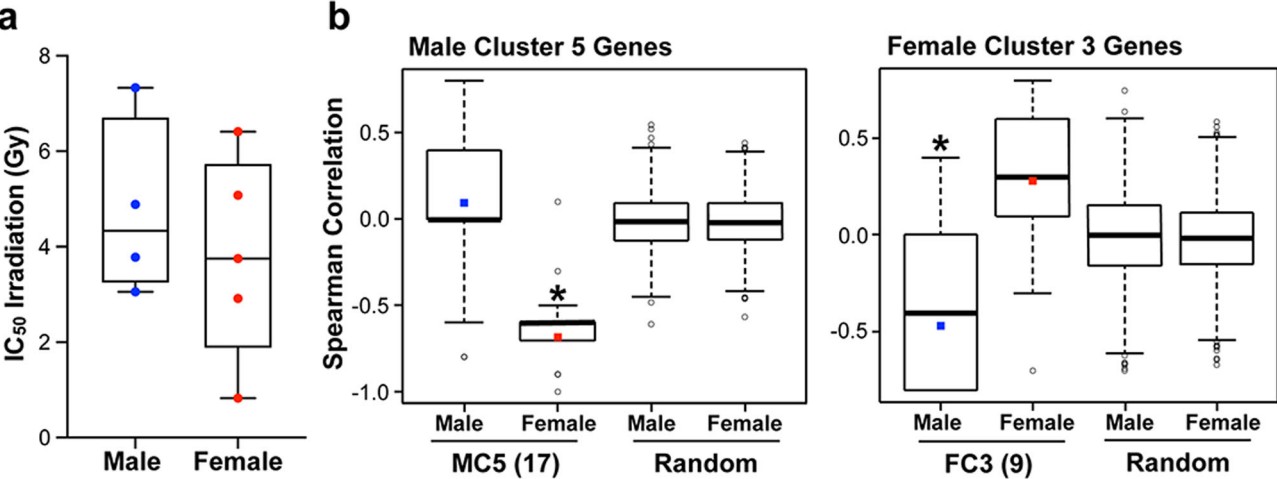

**Fig. 1 Unique pathways contribute to cell-intrinsic radiation sensitivity in male and female human GBM lines. a** Box plots of irradiation $IC_{50}$ values for four male and five female primary human GBM lines (horizontal bar indicates median). **b** Spearman correlation coefficients of irradiation $IC_{50}$ values with expression of MC5 genes (17 genes), FC3 genes (9 genes), or random gene sets of the same size. For MC5 and FC3, box plots represent the distribution of the correlation coefficients for the 17 and 9 genes respectively. The colored square marks the average. For the random gene sets, the box plots represent the distribution of the Olkin-averaged Spearman correlation coefficient for 1000 gene sets of 17 (Male Cluster 5) or 9 (Female Cluster 3) randomly selected genes. MC5: male $p = 0.245$, female $p < 0.001$. FC3: male $p = 0.025$, Female $p = 0.083$.

**Female mouse GBM model astrocytes are more sensitive to radiation treatment**. To better understand the mechanisms underlying radiation response in male and female GBM, we utilized an in vitro mouse model. This model consists of murine astrocytes with loss of function of the tumor suppressors Nf1 and p53 (*Nf1−/− DNp53*)[13]. When implanted intracranially, both male and female cells form tumors that histologically resemble high-grade gliomas, although the frequency of tumor formation differs by cell sex[13]. In addition, this model displays sex differences in gene expression that are concordant with sex differences in human GBM patient gene expression[14]. We performed irradiation dose response curves with male and female *Nf1−/− DNp53* astrocytes and found a significant sex difference (Fig. 2a). Female cells were more sensitive to irradiation, a finding consistent with results from GBM patients that suggest females are more responsive to standard of care therapy (radiation and chemotherapy)[7].

To further assess sex differences in radiation sensitivity, we irradiated male and female *Nf1−/− DNp53* astrocytes with 0, 3, or 9 Gy, then used live cell imaging to track cell growth for 3 days (Fig. 2b). As previously reported[13], in the absence of radiation treatment male cells grew faster than female cells. While both male and female cells showed impairment in growth after irradiation with 9 Gy, only female cells showed a decline in growth after treatment at the lower dose of 3 Gy (Fig. 2b). This suggests that female GBM model astrocytes are more sensitive to low dose irradiation. We next used the colony formation assay to assess clonogenic survival following radiation treatment. Female *Nf1−/− DNp53* astrocytes had a greater decrease in colony counts with radiation treatment than male *Nf1−/− DNp53* astrocytes (Fig. 2c), further supporting the idea that female GBM model cells are more sensitive to irradiation.

Finally, we assessed activation of the DNA damage response (DDR) pathway by irradiating male and female *Nf1−/− DNp53* astrocytes with 3 or 8 Gy and then performing immunofluorescence for γH2AX, a marker of cellular response to DNA double-strand breaks (Fig. 2d, e). We quantified the percent of cells with >10 γH2AX foci at 1, 6, 24, and 48 h after irradiation (Fig. 2e). At 1 h after treatment, essentially 100% of male and female cells were positive. Over time, the percent of positive cells declined significantly more rapidly and more completely in male

compared to female cells. At 48 h, ~10% of male cells treated with 3 or 8 Gy irradiation had >10 γH2AX foci. In contrast, female cells treated with 3 Gy or 8 Gy irradiation still had 25% and 15% positivity for γH2AX foci, respectively, at 48 h. Moreover, while radiation dose had no significant effect on the male cell response, there was a significant difference in female cell response to 3 versus 8 Gy. This suggests that radiation treatment results in more persistent DDR activation in female cells. As γH2AX foci can persist after resolution of double-strand breaks[16], differences in γH2AX staining do not necessarily reflect differences in the efficiency or completeness of DNA repair. The results do, however, indicate that there are sex differences in DDR pathway activation and suggest that persistent DDR activation may contribute to enhanced radiation sensitivity in female cells.

**Cell death and senescence contribute to sex differences in radiation response in mouse GBM model astrocytes.** Cell response to DNA damage can vary. Cells may undergo: (1) transient cell cycle arrest and repair of DNA damage, (2) apoptosis or other forms of cell death, or (3) permanent cell cycle arrest in the form of senescence[17]. We focused on the two mechanisms that result in durable therapeutic responses to cancer therapy—apoptosis/cell death and senescence. To assess apoptosis, we first measured levels of PARP and caspase-3 cleavage following 0, 3, 6, or 8 Gy irradiation. Neither apoptosis marker exhibited radiation dose response 24 h after treatment (Supplementary Fig. 2a). At 5 days post-treatment, caspase-3 (Fig. 3a), but not PARP (Supplementary Fig. 2b), exhibited dose-dependent cleavage in male cells only. These data suggest that apoptosis does not underlie the increased female sensitivity to radiation.

Apoptosis is only one potential mechanism by which cells can die following irradiation. In order to look more quantitatively at cell death, we measured annexin V staining by flow cytometry 24 h and five days after irradiation with 0, 3, 6, or 8 Gy (Fig. 3b, Supplementary Fig. 2c, d). We observed a dose-dependent increase in the annexin V positive fraction at both 24 h (Supplementary Fig. 2d) and 5 days (Fig. 3b) after irradiation. At 24 h there was no significant effect of sex on annexin V staining, while at 5 days females had higher levels of annexin V

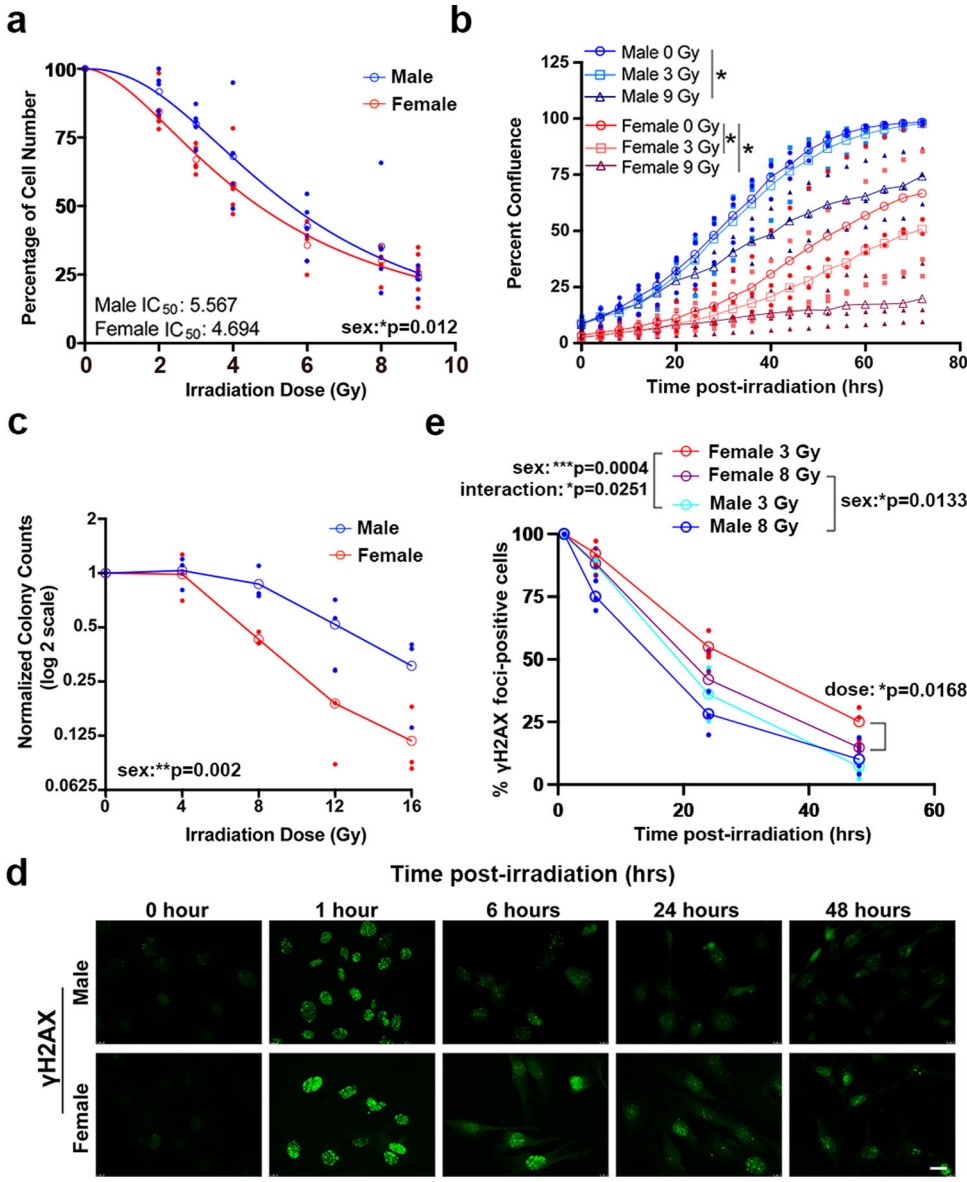

**Fig. 2 Female mouse GBM model astrocytes are more sensitive to radiation treatment. a** Irradiation dose response curves in male and female *Nf1−/− DNp53* astrocytes. Curves represent combined values from five separate established cell lines—each consisting of a corresponding male and female line generated at the same time. Male IC$_{50}$ = 5.567 (95% CI 5.094–6.088), female IC$_{50}$ = 4.694 (95% CI 4.349–5.061). Two-way ANOVA: dose $p < 0.0001$, sex $p = 0.0123$, interaction $p = 0.6034$. Open symbols are the means of the individual data points, which are presented as closed symbols. Lines are non-linear fits to the individual data points ($n = 5$/sex). **b** Male and female *Nf1−/− DNp53* cell growth curves after irradiation with 0, 3, or 9 Gy. Cell growth was tracked using live cell imaging; images were taken every 4 h for 3 days and used to calculate percent confluence over time. Two-way repeated measures ANOVA—male 0 Gy vs male 3 Gy: time $p < 0.0001$, dose $p = 0.3394$, interaction $p = 0.9087$; male 0 Gy vs male 9 Gy: time $p < 0.0001$, dose $p = 0.0325$, interaction $p < 0.0001$; female 0 Gy vs female 3 Gy: time $p < 0.0001$, dose $p = 0.0183$, interaction $p < 0.0001$; female 0 Gy vs female 9 Gy: time $p < 0.0001$, dose $p = 0.05$, interaction $p < 0.0001$. Open symbols are the means of the individual data points, which are presented as closed symbols. Lines are non-linear fits to the individual data points ($n = 3$/sex). **c** Normalized colony counts from the clonogenic assay in male and female *Nf1−/− DNp53* cells 5 days after irradiation with 0, 4, 8, 12, or 16 Gy. Two-way ANOVA: dose $p < 0.0001$, sex $p = 0.002$, interaction $p = 0.111$. Open symbols are the means of the individual data points, which are presented as closed symbols. ($n = 3$/sex). **d** Representative immunofluorescence images of male and female *Nf1−/− DNp53* astrocytes stained for γH2AX 0, 1, 6, 24, and 48 h after irradiation with 3 Gy. Scale bar, 20 μm. **e** Quantification of the percent of cells with >10 γH2AX foci at each timepoint in male and female *Nf1−/− DNp53* astrocytes following irradiation with 3 or 8 Gy. Two-way ANOVA—male 3 Gy vs female 3 Gy: time $p < 0.0001$, sex $p = 0.0004$, interaction $p = 0.0251$; male 8 Gy vs female 8 Gy: time $p < 0.0001$, sex $p = 0.0133$, interaction $p = 0.2784$; female 3 Gy vs female 8 Gy: time $p < 0.0001$, dose $p = 0.0168$, interaction $p = 0.2959$.

positivity. As cell surface exposure of phosphotidylserine is now recognized to be a feature of both apoptotic and non-apoptotic modes of cell death[18], the annexin V staining, together with the cleaved caspase-3 results, suggest that irradiation leads to increased non-apoptotic cell death in females compared to males, which could contribute to greater female radiation sensitivity.

To evaluate the role of senescence in response to irradiation, we stained cells for senescence-associated β-galactosidase (SA-β-gal), the most widely used biomarker of senescence[11,19,20], five days after irradiation with 0, 3, 6, or 8 Gy. Quantification of the percentage of SA-β-gal positive cells revealed a clear, dose-dependent increase in the percentage of senescent cells in both

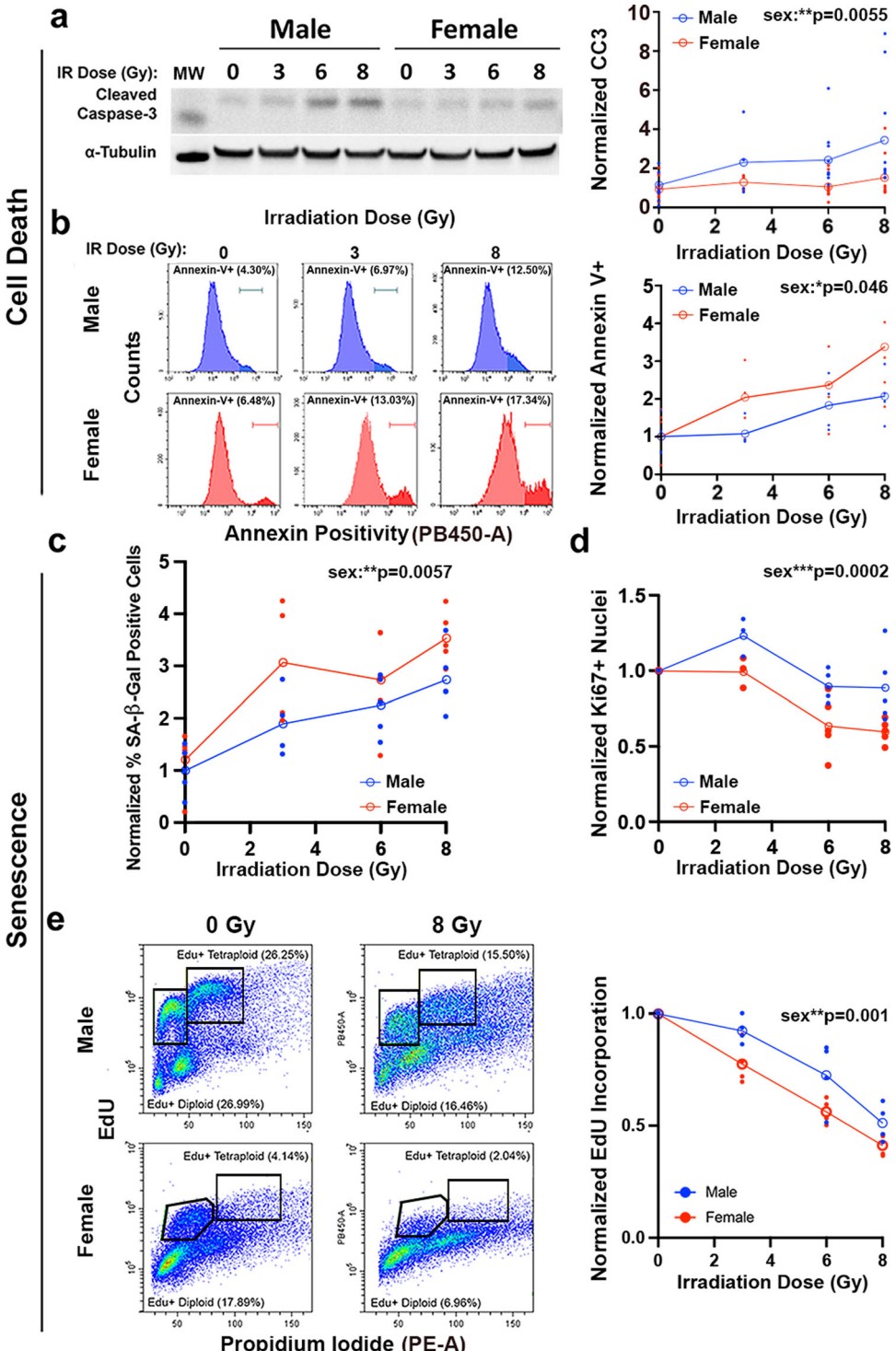

male and female *Nf1−/− DNp53* astrocytes (Fig. 3c). There was a significant sex effect on SA-β-gal positivity, with higher levels of senescent cells observed in females across the range of radiation doses. Consistent with an increase in SA-β-gal, we also observed a change in cell morphology after irradiation[11,19], with a greater percentage of cells appearing enlarged and irregular in shape, as expected for senescence (Supplementary Fig. 3).

To provide further support that senescence was occurring, we performed immunofluorescence for the cell proliferation marker Ki67 5 days after irradiation with 0, 3, 6, or 8 Gy (Fig. 3d, Supplementary Fig. 4a). Ki67 positivity exhibited a significant and dose-dependent decrease after irradiation, again with a significant

effect of sex on the result. As senescence in response to DNA damage involves cell cycle arrest, we directly measured the effects of irradiation on DNA synthesis using flow cytometry to measure EdU incorporation. Concordant with the Ki67 and regrowth after irradiation (Fig. 2b) data, female cells exhibited a continuous decline in EdU incorporation over the entire dose range, while male cells were relatively resistant to 3 Gy and only exhibited substantial arrest at 8 Gy (Fig. 3e, Supplementary Fig. 4b). Finally, we assessed expression levels of two genes that have previously been identified as part of an ionizing radiation-induced senescence (IRIS) signature that was shared across multiple cell types and time points[21]. In agreement with this previous report,

**Fig. 3 Cell death and senescence contribute to sex differences in radiation response in mouse GBM model astrocytes. a** Analysis of radiation-induced caspase-3 cleavage. Representative western blot image of cleaved caspase-3 in male and female $Nf1-/-$ $DNp53$ astrocytes 5 days after irradiation with 0, 3, 6, or 8 Gy; one male and one female cell line are shown. Cleaved caspase-3 (CC3) levels were first normalized to their corresponding α-Tubulin values. Then all CC3/α-Tubulin values were normalized to the corresponding Male 0 Gy condition of the same cell line, which was arbitrarily set at 1. Two-way ANOVA: dose $p = 0.0195$, sex $p = 0.0055$, interaction $p = 0.2085$. Open symbols are the means of the individual data points, which are presented as closed symbols ($n = 9$/sex/dose). Molecular weight markers (MW): Cleaved caspase-3 (15 kDa), α-Tubulin (50 kDa). **b** Analysis of radiation-induced annexin V positivity. Sample histograms are shown for male (blue) and female (red) $Nf1-/-$ $DNp53$ cells irradiated with 0, 3, or 8 Gy and stained for annexin 5 days later. Quantification of annexin V positivity 5 days after irradiation with 0, 3, 6, or 8 Gy. Values were normalized to the corresponding male 0 Gy condition of the same cell line, which was arbitrarily set at 1. Two-way ANOVA: dose $p = 0.0077$, sex $p = 0.0464$, interaction $p = 0.8155$. Open symbols are the means of the individual data points, which are presented as closed symbols ($n = 4$/sex/dose) **c** Quantification of the percentage of SA-β-gal positive cells after irradiation with 0, 3, 6, or 8 Gy. Values were normalized to the corresponding Male 0 Gy condition of the same cell line, which was arbitrarily set at 1. Two-way ANOVA: dose $p < 0.0001$, sex $p = 0.0057$, interaction $p = 0.5356$. Open symbols are the means of the individual data points, which are presented as closed symbols ($n = 4$–$5$/sex/dose). **d** Percent of Ki67 positive cells in male and female $Nf1-/-$ $DNp53$ cultures 5 days after irradiation with 0, 3, 6, or 8 Gy. Values were normalized to the 0 Gy control condition for males and females separately. Two-way ANOVA: dose $p < 0.0001$, sex $p = 0.0002$, interaction $p = 0.08$. Open symbols are the means of the individual data points, which are presented as closed symbols ($n = 3$–$5$/sex/dose). **e** Analysis of EdU incorporation. Example density plots for EdU incorporation in male and female $Nf1-/-$ $DNp53$ astrocytes under control conditions or 24 h after irradiation with 8 Gy. These cultures have both 2 N and 4 N populations and S-phase Edu incorporation is indicated by black gates. Quantification of combined 2 N and 4 N EdU incorporation 24 h after irradiation with 0, 3, 6, or 8 Gy. Values were normalized to the 0 Gy control condition for males and females separately. Two-way ANOVA: dose $p < 0.0001$, sex $p = 0.001$, interaction $p = 0.177$. Open symbols are the means of the individual data points, which are presented as closed symbols ($n = 4$/sex/dose).

we observed an increase in expression of $Ccnd1$ (Cyclin D1) and a decrease in expression of $Cdkn1b$ (p27) 5 days after irradiation in both male and female $Nf1-/-$ $DNp53$ astrocytes (Supplementary Fig. 4c, d). Together, these findings indicate that cell cycle arrest and senescence are a central component of the response to irradiation in $Nf1-/-$ $DNp53$ astrocytes. Sex had a significant effect across multiple measures of the senescence response, including SA-β-gal, Ki67, and EdU incorporation, indicating that female cell sex is associated with greater senescence in response to irradiation compared to male cells.

Together these results demonstrate that at 5 days after irradiation female cells have increased levels of cell death and senescence compared to male cells. Both of these processes could contribute to sex differences in radiation sensitivity. Intriguingly, while we saw no sex differences in cell death 24 h after irradiation, female cells did undergo greater cell cycle arrest compared to males. This suggests there may be sex differences in the early activation of pathways that lead to arrest, and eventually senescence, following irradiation.

**Expression of p21 24 h after irradiation correlates with the senescence response observed at 5 days.** Two critical pathways for the induction of cell cycle arrest and cellular senescence are the p16[INK4A]/Rb and p53/p21[WAF1/Cip1] pathways[10,11]. The $Nf1-/-$ $DNp53$ astrocytes express a dominant negative p53, which has been demonstrated to abrogate canonical wild-type p53 transcriptional activity[13,22]. Nonetheless, we have shown that these cells retain the ability to upregulate p21 in response to DNA damage[14], presumably through p53-independent mechanisms[23]. To better understand the contributions of the p16[INK4A]/Rb and p53/p21[WAF1/Cip1] pathways to senescence induction following irradiation, we used qPCR to measure expression levels of $Cdkn2a$ (p16) and $Cdkn1a$ (p21) 24 h after treatment of cells with 0, 6, or 9 Gy (for nomenclature clarification, we will use the gene name ($Cdkn2a$, $Cdkn1a$) when referring to mRNA expression levels, and the protein name (p16, p21) when referring to protein expression levels). We then correlated these values with the percentage of cells that were SA-β-gal positive at 5 days. Because our lab has previously identified sex differences in the regulation of both p16 and p21 in $Nf1-/-$ $DNp53$ astrocytes[14], we looked at the relationship between these two measures and senescence in males and females separately. $Cdkn2a$ levels 24 h after irradiation showed no significant correlation with the senescence response at

5 days in either males or females (Fig. 4a, Supplementary Table 3). In contrast, levels of $Cdkn1a$ significantly correlated with the percent of SA-β-gal positive cells in both males ($r = 0.59$) and females ($r = 0.79$) (Fig. 4b, Supplementary Table 3). This suggests that early upregulation of p21, but not p16, contributes to the senescence response following irradiation in mouse GBM astrocytes. This is consistent with previous research from our lab, which found that in response to treatment with the DNA damaging agent etoposide, levels of p21, but not p16, increased in $Nf1-/-$ $DNp53$ cells[14]. To confirm that increased p21 expression was independent of the wild-type p53 alleles in our cell model, we directly measured the endogenous p53 levels 24 h and five days following irradiation with 0, 3, 6, or 8 Gy. Whereas p21 expression was induced by irradiation in male and female $Nf1-/-$ $DNp53$ cells, there was no concomitant induction of p53 (Supplementary Fig. 5).

Downstream of p21 is the cyclin-dependent kinase Cdk2, and it is through inhibition of Cdk2 that p21 primarily acts to maintain Rb in a hypo-phosphorylated state[24,25], a critical step in cell cycle arrest and senescence induction[11]. In addition, it has been reported that the p21/Cdk2 ratio is the primary determinant of the senescent fate decision in human fibroblasts following irradiation[26]. To assess whether the p21/Cdk2 ratio may similarly influence the induction of senescence in our mouse GBM model cells, we measured levels of $Cdk2$ mRNA 24 h after irradiation and correlated the $Cdkn1a$/$Cdk2$ ratio with the percent of SA-β-gal positive cells at 5 days (Fig. 4c, Supplementary Table 3). We found that the $Cdkn1a$/$Cdk2$ ratio significantly correlated with SA-β-gal positivity in both males ($r = 0.53$) and females ($r = 0.80$).

**p21 differentially correlates with SA-β-gal positivity in male and female mouse GBM model astrocytes.** We next sought to determine whether the correlation between the p21/Cdk2 ratio and senescence was also observed at the protein level, and whether this relationship was maintained at later timepoints, once senescence was established. To this end, we measured p21 and Cdk2 levels by western blot, 24 h and 5 days after irradiation with 0, 3, 6, and 8 Gy (Fig. 4d–f). Surprisingly, while p21/Cdk2 protein levels at 24 h significantly correlated with SA-β-gal positivity in females ($r = 0.52$, $p = 0.0029$), this was not the case for males ($r = 0.30$, $p = 0.997$) (Fig. 4e, Supplementary Table 3). Even more striking, while p21/Cdk2 protein levels at 5 days did correlate

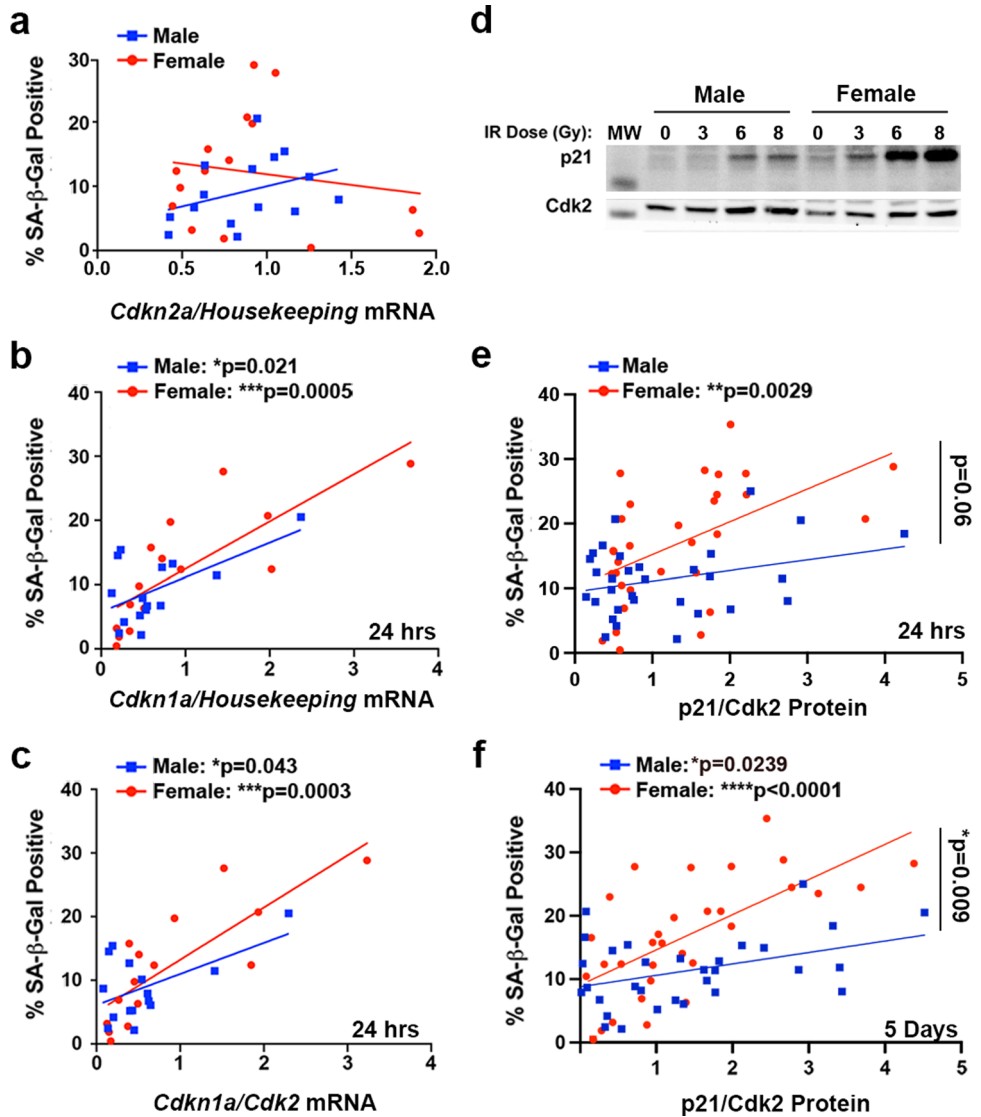

**Fig. 4 Expression of p21 differentially correlates with SA-β-gal positivity in male and female mouse GBM model astrocytes. a** Correlation between *Cdkn2a* (p16) mRNA levels at 24 h after irradiation with 0, 6, or 9 Gy and the percentage of SA-β-gal positive cells at 5 days after irradiation in male and female *Nf1−/− DNp53* astrocytes (*n* = 15/sex). **b** Correlation between *Cdkn1a* (p21) mRNA levels at 24 h after irradiation with 0, 6, or 9 Gy and the percentage of SA-β-gal positive cells at 5 days after irradiation in male and female *Nf1−/− DNp53* astrocytes. Male correlation: *r* = 0.59, *p* = 0.0212; Female correlation: *r* = 0.79, *p* = 0.0005 (*n* = 15/sex). **c** Correlation between the *Cdkn1a/Cdk2* mRNA ratio at 24 h after irradiation with 0, 6, or 9 Gy and the percentage of SA-β-gal positive cells at 5 days after irradiation in male and female *Nf1−/− DNp53* astrocytes. Male correlation: *r* = 0.53, *p* = 0.0427; Female correlation: *r* = 0.80, *p* = 0.0003 (*n* = 15/sex). **d** Representative western blot images of p21 and Cdk2 5 days post-irradiation with 0, 3, 6, or 8 Gy. One male and one female *Nf1−/− DNp53* cell line are shown. Molecular weight markers (MW): p21 (15 kDa), Cdk2 (30 kDa). **e** Correlation between the p21/Cdk2 protein ratio at 24 h after irradiation with 0, 3, 6, or 8 Gy and the percentage of SA-β-gal positive cells at 5 days after irradiation in male and female *Nf1−/− DNp53* astrocytes. Male correlation: *r* = 0.30, *p* = 0.0997, female correlation: *r* = 0.52, *p* = 0.0029, male slope vs female slope: *p* = 0.0627—indicated by bracket (*n* = 31/sex). **f** Correlation between the p21/Cdk2 protein ratio 5 days after irradiation with 0, 3, 6, or 8 Gy and the percentage of SA-β-gal positive cells at 5 days after irradiation in male and female *Nf1−/− DNp53* astrocytes. Male correlation: *r* = 0.40, *p* = 0.0239, female correlation: *r* = 0.65, *p* < 0.0001, male slope vs female slope: *p* = 0.009—indicated by bracket (*n* = 31/sex).

with the percentage of SA-β-gal positive cells in both males (*r* = 0.40, *p* = 0.0239) and females (*r* = 0.65, *P* < 0.0001), it was with significantly different slopes (*p* = 0.009) (Fig. 4f, Supplementary Table 3). Thus, the same level of p21/Cdk2 expression at 5 days corresponded to higher levels of senescence in females than in males, as measured by SA-β-gal. This suggests that not only does the p21/Cdk2 ratio play a role in the maintenance of senescence following irradiation, but that there may be a sex difference in sensitivity to p21/Cdk2 levels and/or that p21 may be playing a greater role in irradiation-induced senescence in female GBM cells than in male GBM cells.

**Wild-type mouse astrocytes exhibit sex differences in the relationship between p21 and SA-β-gal**. We wondered whether the sex difference in the relationship between p21 and SA-β-gal that we observed in our mouse GBM model represents a more fundamental sex difference, present in normal astrocytes, or is unique to transformed cells and the loss of p53 function. To address this question, we isolated astrocytes from the cortex of male and female postnatal day 1 C57Bl6 mouse pups. Astrocytes from each pup were cultured independently and split to allow for corresponding measures of SA-β-gal and collection of RNA or protein. We irradiated these wild-type (WT) astrocytes with 0 or

10 Gy, then collected RNA or protein 24 h later, and stained for SA-β-gal at 7 days (Fig. 5a). The irradiation dose and time until SA-β-gal measurement were increased to adjust for the much slower division rate of wildtype astrocytes compared to *Nf1−/− DNp53* astrocytes. Even in the untreated condition, wildtype astrocytes exhibited measurable levels of senescence, reflective of their limited replicative potential (Fig. 5a). Quantification of the percentage of SA-β-gal positive cells confirmed a significant increase following irradiation in both male and female wildtype astrocytes (Fig. 5b). To confirm that cells were undergoing $G_1$ arrest, consistent with senescence, we performed immuno-fluorescence for geminin, which is absent during $G_1$ and accumulates during S, $G_2$, and M phases of the cell cycle. We counted the numbers of geminin-positive male and female cells per high-powered field (hpf) 7 days post-treatment. As expected, the number of geminin-positive cells significantly decreased in both male and female astrocytes in response to irradiation (Fig. 5c).

We used qPCR to measure expression levels of *Cdkn2a* (p16) and *Cdkn1a* (p21) mRNA at 24 h after irradiation, then correlated these measures with the corresponding SA-β-gal percentage. As expected, *Cdkn2a* levels did not significantly correlate with SA-β-gal positivity in either male or female WT astrocytes (Fig. 5d, Supplementary Table 3), confirming that p16 upregulation is not a primary driver of senescence induction in astrocytes following irradiation. In contrast, *Cdkn1a* levels did significantly correlate with the percent of SA-β-gal positive cells at 7 days, however, only in females (female $r = 0.79$, male $r = 0.28$) (Fig. 5e, Supplementary Table 3). When we looked at the relationship between the *Cdkn1a/Cdk2* ratio and SA-β-gal, we once again found a significant correlation in female WT astrocytes ($r = 0.94$), but only a non-significant mild correlation in male WT astrocytes ($r = 0.39$) (Fig. 5f, Supplementary Table 3). As with the *Nf1−/− DNp53* astrocytes, the female WT astrocytes had a steeper slope for both *Cdkn1a* and *Cdkn1a/Cdk2* vs SA-β-gal compared to males, although this did not reach statistical significance.

To determine whether the sex difference in the relationship between p21 expression and SA-β-gal was maintained at the protein level, we measured p21 and Cdk2 protein levels by western blot (Fig. 5g), then correlated these with the percent of SA-β-gal positive cells at 7 days. Both p21 alone, and the p21/Cdk2 ratio, significantly correlated with senescence in female WT astrocytes ($r = 0.67$ and $r = 0.53$, respectively), but not male WT astrocytes (Fig. 5h, i, Supplementary Table 3). These results suggest that p21 may be playing a greater role in senescence induction in female astrocytes than in male astrocytes following irradiation, and that this is occurring regardless of whether the cells are transformed or not.

**Sex differences in senescence are observed in wild-type astrocytes with advanced population doublings.** In initial studies to assess the effects of treatment-induced senescence in wild-type astrocytes, we observed that with successive population doubling levels (PDLs) a baseline sex difference in senescence began to emerge – with females having higher percentages of SA-β-gal positive cells than males in the untreated condition. To assess this more directly, we cultured male and female wildtype astrocytes and stained for SA-β-gal when the cells were at either low (5–7) versus high (12–14) PDLs (Supplementary Fig. 6a). At 12–14 PDLs, wild-type astrocytes showed a dramatic decrease in growth, with the majority of cells adopting an enlarged, flattened, irregular morphology, and were unable to be passaged further.

Male and female wild-type astrocytes proliferate at equal rates, and thus we were able to make ready comparisons between the senescent cell fractions as a function of PDLs in male and female cultures. At low (5–7) PDLs, the ratio of SA-β-gal positive cells

(female/male) was ~1, indicating equivalent levels of senescent cells in male and female astrocyte cultures (Supplementary Fig. 6b). At higher (12–14) PDLs, this ratio was significantly increased, indicating higher percentages of senescent cells in female cultures. Thus, female mouse astrocytes undergo senescence more frequently than male mouse astrocytes with increased PDLs. Whether the higher rates of senescence in female wild-type astrocyte cultures reflect sex differences in the propensity to undergo replicative senescence or in the sensitivity to oxidative damage, resulting from culture at 20% oxygen[27], remains to be determined. However, this suggests that sex differences may extend to other senescence paradigms and phenotypes, and highlights the need to study these in both sexes separately.

**Primary human GBM lines exhibit sex differences in the relationship between p21/Cdk2 and SA-β-gal.** We next assessed whether sex differences in the role of p21 in irradiation-induced senescence extended to human GBM. For this purpose, we utilized the same primary human GBM lines used to correlate irradiation sensitivity with expression of the MC5 and FC3 gene sets. We irradiated each male and female line with 0 or 6 Gy and stained for SA-β-gal 5 days later. The 6 Gy dose was chosen based on the human GBM irradiation dose response curves; it was above the mean $IC_{50}$ value for both sexes, but lower than the dose at which most lines showed a plateau in response (Supplementary Fig. 1). There was a significant increase in the percent of SA-β-gal positive cells following irradiation in both male and female human GBM (Fig. 6a).

Using protein samples collected 24 h and 5 days after irradiation, we measured levels of p21 and Cdk2 by western blot (Fig. 6b). At 24 h, both p21 alone (Fig. 6c) and the p21/Cdk2 ratio (Fig. 6d) significantly correlated with the percent of SA-β-gal positive cells at 5 days in female human GBM lines ($r = 0.85$ for both), but only moderately correlated in male human GBM lines ($r = 0.23$ and $r = 0.54$, respectively) (Supplementary Table 3). This same pattern was observed at 5 days, with p21 and p21/Cdk2 significantly correlating with SA-β-gal for female ($r = 0.83$ and $r = 0.84$, respectively) but not male lines ($r = -0.09$ and $r = −0.27$, respectively) (Fig. 6e, f, Supplementary Table 3). Given the small number of human GBM specimens, we considered whether these correlations were being driven by outlier versus leverage points[28]. We concluded that these were leverage points as they were characterized by having "extreme" x and y values that followed the overall linear relationship. In addition, they were consistent with what we observed in the wild-type and transformed murine models. Thus, in both mouse and human, transformed and wild-type astrocytes, the relationship between p21 and senescence, as measured by SA-β-gal, differs between males and females.

**Knockdown of p21 abrogates sex differences in senescence and DNA damage response in irradiated mouse GBM model astrocytes.** To further test the role of p21 in senescence in irradiated male and female cells, we utilized an *Nf1−/− DNp53* p21 knockdown line previously developed in our lab using CRISPR/Cas9[14]. We confirmed knockdown by irradiating Cas9 and p21 KD cells with 0 or 8 Gy and measuring levels of p21 protein 24 h later. Both male and female p21 KD lines had a strong reduction in p21 levels—less than 30% of control levels (Supplementary Fig. 7a, b). We then irradiated male and female Cas9 and p21 KD cells with 0, 3, 6, or 8 Gy and stained for SA-β-gal 5 days later (Fig. 7a). Once again, in control Cas9 cells, females exhibited a greater dose-dependent senescence response than males (Fig. 7b). The male response was unaffected by p21 knockdown (Fig. 7c). In contrast, p21 knockdown in female cells resulted in a significant

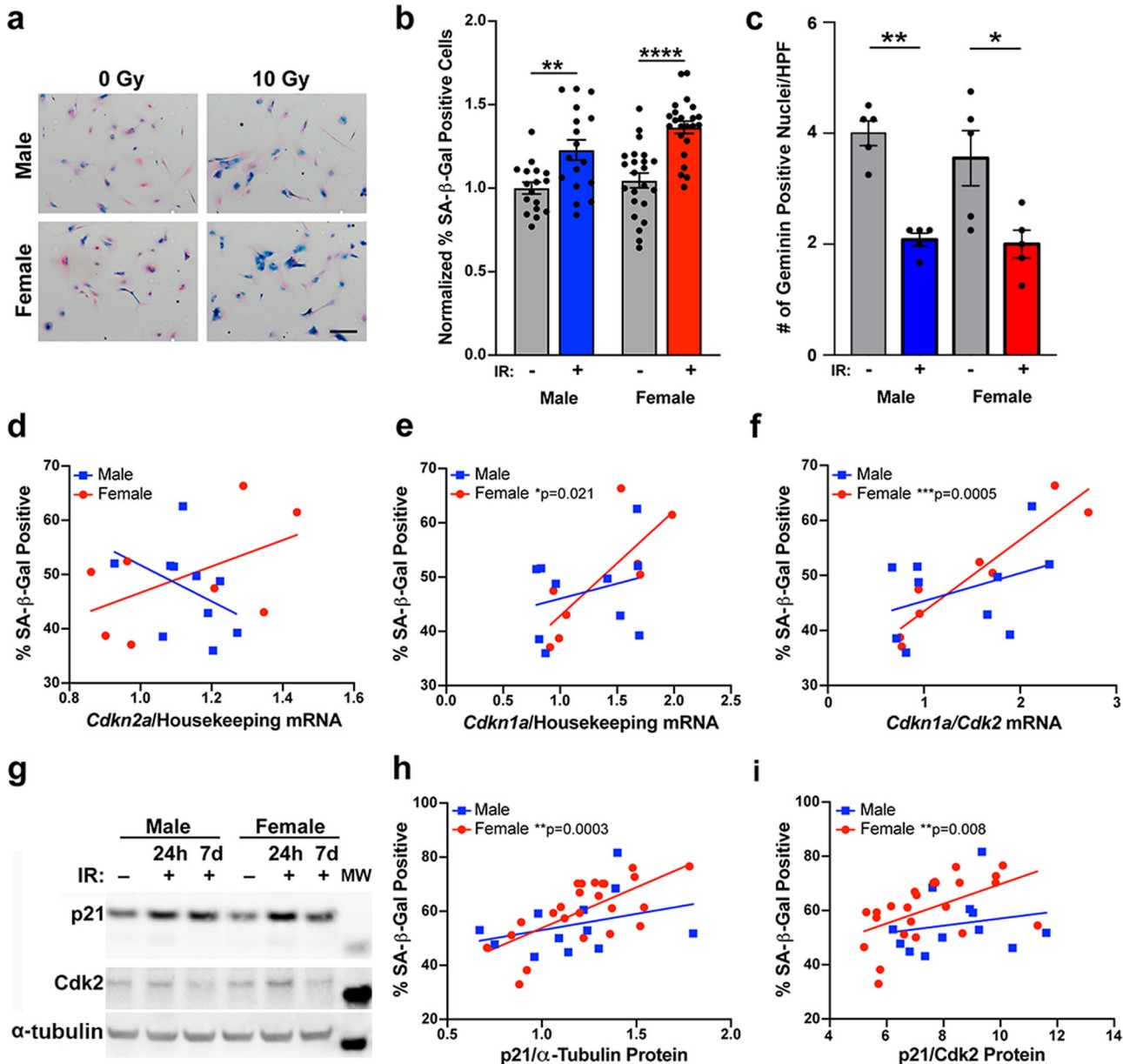

**Fig. 5 Wild-type mouse astrocytes exhibit sex differences in the relationship between p21 and SA-β-gal positivity. a** Example images of wildtype mouse astrocytes stained for SA-β-gal seven days after irradiation with 0 or 10 Gy, followed by counterstaining with nuclear fast red. Scale bar, 150 μm. **b** Quantification of the percentage of SA-β-gal positive cells. Results are pooled from 3 separate experiments, representing a total of 17 male and 23 female pups across 7 litters. Values were normalized to the Male 0 Gy condition of the same experiment, which was arbitrarily set at 1. Two-way ANOVA: dose $p < 0.0001$, sex $p = 0.0455$, interaction $p = 0.3676$. **$p < 0.01$, ****$p < 0.0001$ as indicated by bracket. Data are means +/− SEM ($n = 17-23$/sex/ condition). **c** Number of Geminin positive nuclei per high-powered field. Shown are the means +/− SEM and each of the individual replicate points ($n = 5$/ sex/condition). Two-way ANOVA: dose $p < 0.0001$, sex $p = 0.3948$, interaction $p = 0.5561$. *$p = 0.0145$ and **$p = 0.0024$ as indicated by bracket. **d** Correlation between *Cdkn2a* (p16) mRNA levels at 24 h after irradiation with 0 or 10 Gy and the percentage of SA-β-gal positive cells at 7 days after irradiation in male and female wildtype astrocytes ($n = 10$ male/8 female). **e** Correlation between *Cdkn1a* (p21) mRNA levels at 24 h after irradiation with 0 or 10 Gy and the percentage of SA-β-gal positive cells at 7 days after irradiation in male and female wildtype astrocytes. Male correlation: $r = 0.28$, $p = 0.441$, female correlation: $r = 0.79$, $p = 0.0207$ ($n = 10$ male/8 female). **f** Correlation between the *Cdkn1a/Cdk2* mRNA ratio at 24 h after irradiation with 0 or 10 Gy and the percentage of SA-β-gal positive cells at 7 days after irradiation in male and female wild-type astrocytes. Male correlation: $r = 0.39$, $p = 0.2608$, Female correlation: $r = 0.94$ $p = 0.0005$ ($n = 10$ male/8 female). **g** Representative western blot images of p21 and Cdk2 in non-irradiated and irradiated male and female wild-type astrocytes at 24 h, and irradiated male and female wild-type astrocytes at 7 days. Molecular weight markers (MW): p21 (15 kDa), Cdk2 (30 kDa), α-Tubulin (50 kDa). **h** Correlation between p21 protein levels at 24 h after irradiation with 0 or 10 Gy and the percentage of SA-β-gal positive cells at 7 days after irradiation in male and female wild-type astrocytes. Male correlation: $r = 0.33$, $p = 0.2909$, female correlation: $r = 0.67$, $p = 0.0003$ ($n = 12$ male/24 female). **i** Correlation between the p21/Cdk2 protein ratio at 24 h after irradiation with 0 or 10 Gy and the percentage of SA-β-gal positive cells at 7 days after irradiation in male and female wildtype astrocytes. Male correlation: $r = 0.20$, $p = 0.5418$, Female correlation: $r = 0.53$, $p = 0.0081$ ($n = 12$ male/24 female).

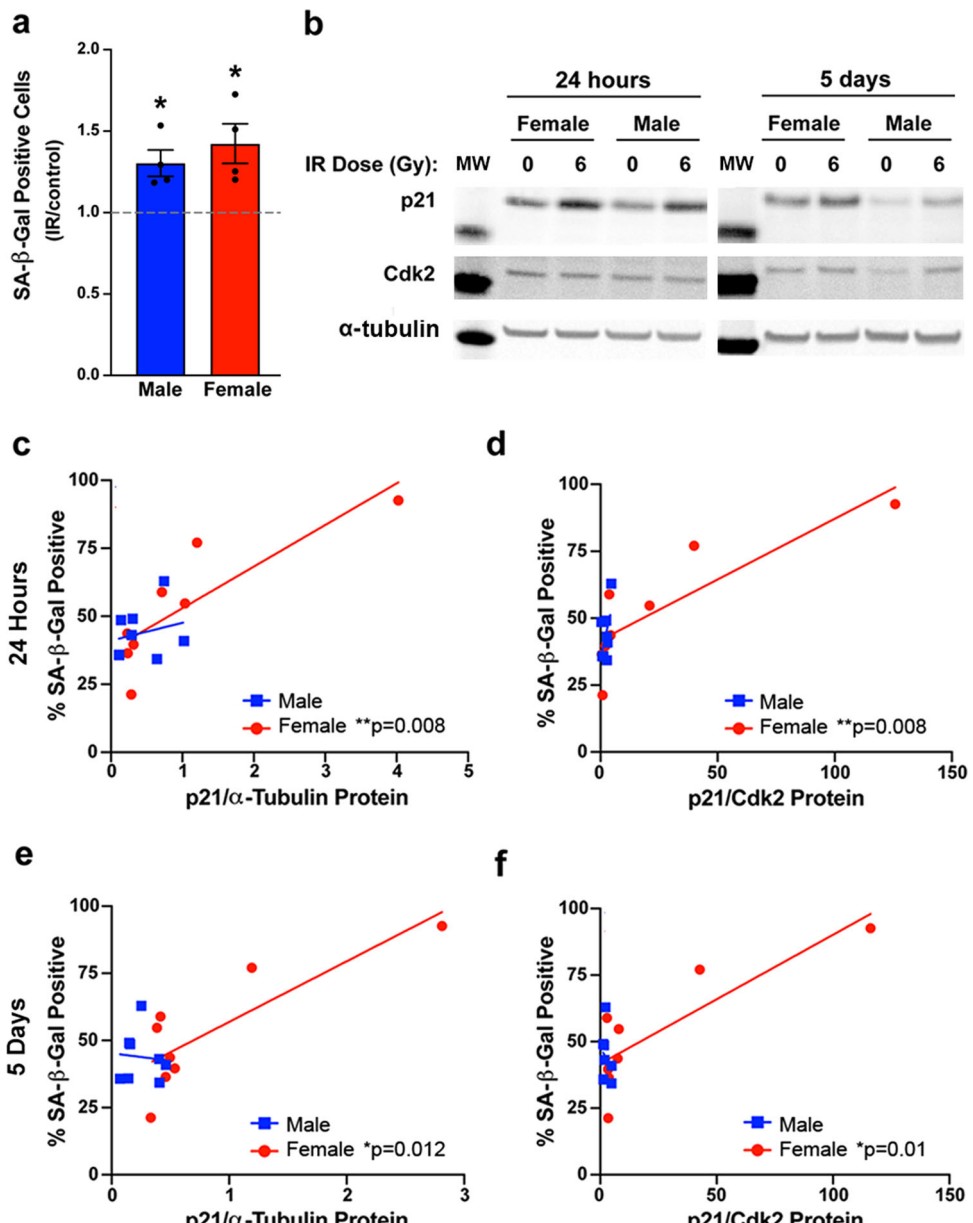

**Fig. 6 Primary human GBM lines exhibit sex differences in the relationship between p21 and SA-β-gal positivity. a** Fold change (irradiated/control) in the percentage of SA-β-gal positive cells 5 days after irradiation with 0 or 6 Gy in four male and four female primary human GBM lines. *$p = 0.0329$ (male) and *$p = 0.0401$ (female). **b** Representative western blot images of p21 and Cdk2 24 h and 5 days after irradiation with 0 or 6 Gy in male and female human GBM lines. Molecular weight markers (MW): p21 (15 kDa), Cdk2 (30 kDa), α-Tubulin (50 kDa). **c** Correlation between p21 protein levels at 24 h after irradiation with 0 or 6 Gy and the percentage of SA-β-gal positive cells at 5 days after irradiation in male and female human GBM lines. Male correlation: $r = 0.23$, $p = 0.5783$, female correlation: $r = 0.85$, $p = 0.008$ ($n = 8$/sex). **d** Correlation between the p21/Cdk2 protein ratio at 24 h after irradiation with 0 or 6 Gy and the percentage of SA-β-gal positive cells at 5 days after irradiation in male and female human GBM lines. Male correlation: $r = 0.54$, $p = 0.1649$, female correlation: $r = 0.85$, $p = 0.0081$ ($n = 8$/sex). **e** Correlation between p21 protein levels at 5 days after irradiation with 0 or 6 Gy and the percentage of SA-β-gal positive cells at 5 days after irradiation in male and female human GBM lines. Male correlation: $r = -0.09$, $p = 0.8408$, female correlation: $r = 0.83$, $p = 0.0115$ ($n = 8$/sex). **f** Correlation between the p21/Cdk2 protein ratio at 5 days after irradiation with 0 or 6 Gy and the percentage of SA-β-gal positive cells at 5 days after irradiation in male and female human GBM lines. Male correlation $r = -0.27$, $p = 0.5179$, female correlation: $r = 0.84$, $p = 0.0099$ ($n = 8$/sex).

reduction in the percent of SA-β-gal positive cells as a function of radiation dose, to levels comparable to those of male cells (Fig. 7d).

To determine whether p21 knockdown would also exhibit a sex-biased effect on DDR, we quantified γH2AX foci resolution in Cas9 control and p21 KD $Nf1-/-$ $DNp53$ astrocytes following irradiation with either 3 or 8 Gy. Consistent with our findings in

Fig. 2e, female Cas9 control cells exhibited sustained γH2AX foci compared to male Cas9 control cells (Fig. 7e). Similar to the effects of p21 knockdown on senescence, the kinetics and magnitude of γH2AX foci resolution following irradiation with 3 or 8 Gy was unaffected by p21 status in male cells (Fig. 7f). In contrast, the kinetics and magnitude of γH2AX foci resolution after treatment with 8 Gy were significantly increased in female

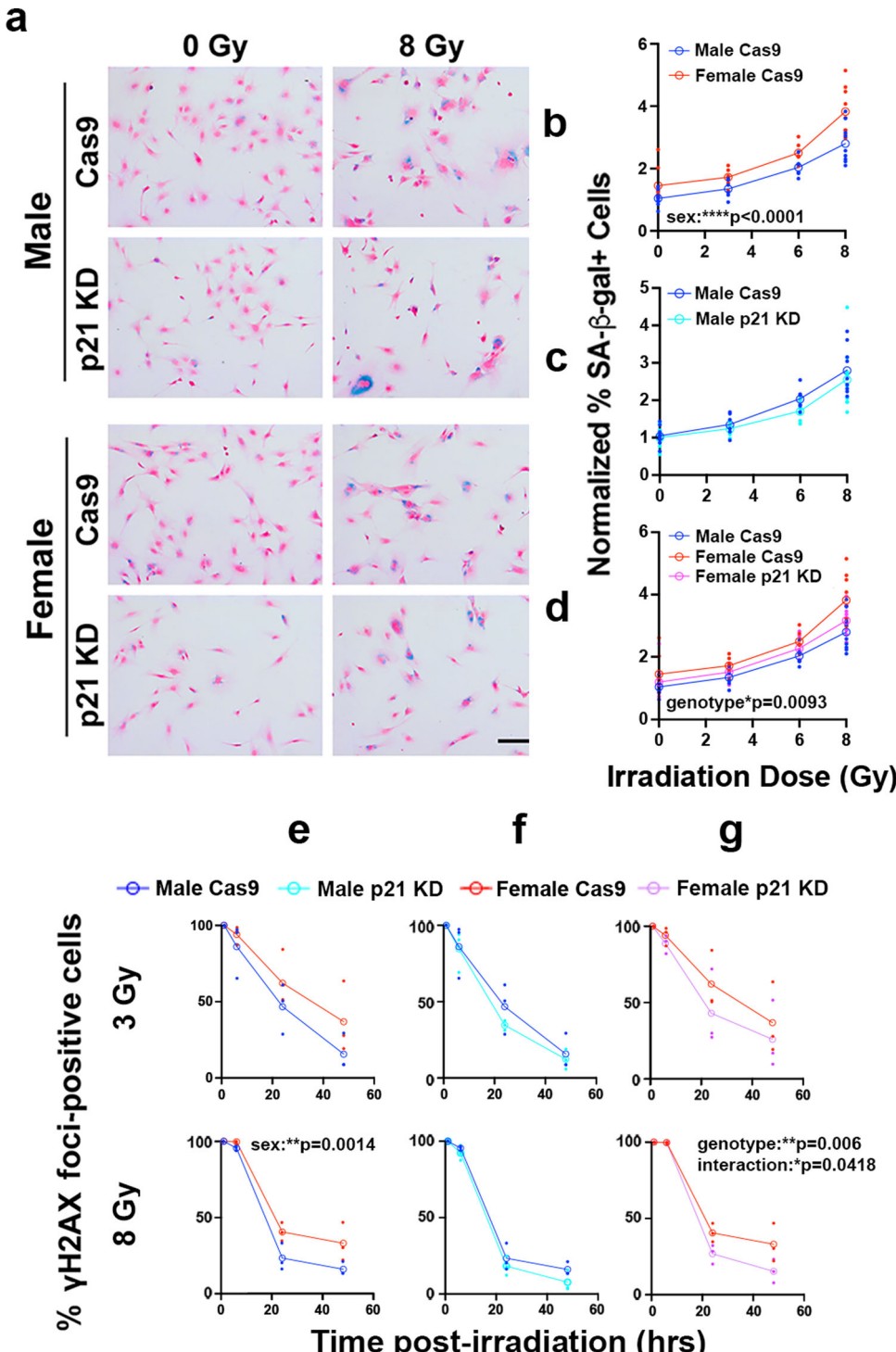

cells by p21 KD (Fig. 7g) to male comparable levels. There was a trend towards an effect of p21 KD in female cells irradiated with 3 Gy, however this did not reach statistical significance. Together, these data support the idea that p21 expression contributes significantly to the levels of senescence after irradiation in females, but not in males. Notably, despite the decrease in SA-β-gal in female p21 KD cells, they still retain a significant senescence response, and in fact decrease only to the levels of senescence in male cells. This suggests that additional pathways, beyond p21, are contributing to senescence after irradiation in both male and female cells.

**The four core genotypes mouse model can be used to interrogate the developmental mechanisms underlying sex differences.** Multiple mechanisms can potentially underlie an observed sex difference: (1) acute differences in circulating gonadal hormone levels, (2) organizational/epigenetic effects of gonadal hormone exposure in utero, and (3) differences in the expression levels of genes on the X and Y chromosomes[29]. The sex difference in the relationship between p21 and SA-β-gal that we observe is present in vitro, with male and female cells grown in identical media. Thus, it is unlikely that the differences in p21 are driven by effects of circulating estrogen or testosterone. To investigate

**Fig. 7 p21 knockdown abrogates sex differences in senescence and DNA damage response in irradiated mouse GBM model astrocytes. a** Example images of $Nf1-/-$ $DNp53$ Cas9 control and p21 knockdown (KD) male and female astrocytes stained for SA-β-gal 5 days after irradiation with 0 or 8 Gy, followed by counterstaining with nuclear fast red. Scale bar, 150 μm. **b–d** Quantification of the percentage of SA-β-gal positive cells 5 days post-irradiation with 0, 3, 6, or 8 Gy. Values were normalized to the corresponding Male 0 Gy condition of the same cell line, which was arbitrarily set at 1. **b** Male vs female Cas9 control cells—two-way ANOVA: dose $p < 0.0001$, sex $p < 0.0001$, interaction $p = 0.2093$. **c** Male Cas9 vs male p21 KD - two-way ANOVA: dose $p < 0.0001$, genotype $p = 0.1459$, interaction $p = 0.8529$; **d** Female Cas9 vs female p21 KD: two-way ANOVA: dose $p < 0.0001$, genotype $p = 0.0093$, interaction $p = 0.5635$. Male Cas9 curve (blue) is shown for comparison. Open symbols are the means of the individual data points, which are presented as closed symbols (**b–d** $n = 6$–9/sex/dose). **e–g** Quantification of the percent of $Nf1-/-$ $DNp53$ Cas9 control versus p21 knockdown (KD) cells with >10 γH2AX foci at 1, 6, 24, and 48 h following irradiation with 3 Gy (top row) or 8 Gy (bottom row). Two-way ANOVAs: **e** Male Cas9 versus female Cas9 at 3 Gy: time: $p < 0.0001$, sex: $p = 0.079$, interaction: $p = 0.625$; male Cas9 versus female Cas9 at 8 Gy: time: $p < 0.0001$, sex: $p = 0.0014$, interaction: $p = 0.0524$. **f** Male Cas9 versus p21 KD at 3 Gy: time $p < 0.0001$, genotype: $p = 0.356$, interaction: $p = 0.775$; male Cas9 versus p21 KD at 8 Gy: time $p < 0.0001$, genotype: $p = 0.0645$, interaction: $p = 5773$ **g** female Cas9 versus p21 KD at 3 Gy: time $p < 0.0001$, genotype: $p = 0.205$, interaction: $p = 0.774$; female Cas9 versus p21 KD at 8 Gy: time $p < 0.0001$, genotype: $p = 0.006$, interaction: $p = 0.0418$. Open symbols are the means of the individual data points, which are presented as closed symbols (**e–g** $n = 3$/sex/timepoint).

whether organizational effects or sex chromosomes are responsible, we utilized a transgenic mouse model known as the Four Core Genotypes (FCG) model[30]. In the FCG model, the $Sry$ gene, which encodes the testis-determining factor, is deleted from the Y chromosome and inserted onto an autosome, allowing for the separation of gonadal and chromosomal sex. Crossing XY⁻/Sry+ males with normal XX females results in mice of four genotypes: XY⁻/Sry+ (XY with testes—normal male), XY⁻/Sry− (XY with ovaries), XX/Sry+ (XX with testes), and XX/Sry− (XX with ovaries —normal female) (Fig. 8a). Mice that inherit the $Sry$ gene, regardless of chromosomal sex, develop testes and are exposed to masculinizing levels of gonadal hormones during *in utero* development. Mice that lack the $Sry$ gene develop ovaries and display phenotypes associated with a feminized brain[30]. We crossed FCG XY⁻/Sry+ mice with Cas9 expressing female (XX) mice and isolated astrocytes from the postnatal day 1 pups. We then used CRISPR/Cas9 to delete Nf1 and p53 from these astrocytes, mimicking our $Nf1-/-$ $DNp53$ GBM model (Supplementary Fig. 8).

**Gonadal sex patterns the relationship between p21 and SA-β-gal positivity**. Using our newly developed FCG GBM model, we irradiated the cells with 0 or 8 Gy and then stained for SA-β-gal 5 days later (Fig. 8b). The percentage of SA-β-gal positive cells increased following irradiation in all four genotypes (Fig. 8c), although due to high variability, this did not reach significance in the XY+ genotype. Using qPCR, we measured levels of $Cdkn1a$ (p21) mRNA expression 24 h after irradiation and correlated this with the percent of SA-β-gal positive cells at 5 days (Fig. 8d, left, Supplementary Table 3). To better evaluate whether sex chromosomes or gonadal sex determines the relationship between p21 and SA-β-gal, we grouped the genotypes based on these factors and then compared XY vs XX (Fig. 8d, center, Supplementary Table 3) and Sry+ vs Sry− (Fig. 8d, right, Supplementary Table 3). When the genotypes were grouped based on sex chromosomes, $Cdkn1a$ did not significantly correlate with SA-β-gal in either XY or XX FCG GBM model cells. However, when the genotypes were grouped based on gonadal sex, $Cdkn1a$ expression significantly correlated with SA-β-gal positivity in Sry− cells ($r = 0.91$), but not Sry+ cells ($r = 0.13$).

We next measured $Cdk2$ levels by qPCR and correlated the $Cdkn1a/Cdk2$ ratio with SA-β-gal percentage (Fig. 8e, left, Supplementary Table 3). When we grouped the genotypes based on sex chromosomes, we again observed that the $Cdkn1a/Cdk2$ ratio did not correlate with SA-β-gal in either XY or XX cells (Fig. 8e, center, Supplementary Table 3). In contrast, when we grouped the genotypes based on gonadal sex, the $Cdkn1a/Cdk2$ ratio significantly correlated with SA-β-gal in Sry− cells ($r = 0.91$), but not Sry+ cells ($r = -0.22$) (Fig. 8e, right,

Supplementary Table 3). In addition, when we compared the slopes of the lines for Sry+ and Sry−, they were significantly different (Supplementary Table 3). Thus, FCG GBM astrocytes isolated from mice that were gonadally female had a significant relationship between p21 and senescence, as measured by SA-β-gal. This suggests that gonadal sex, and the epigenetic effects of in utero gonadal hormones, patterns the role of p21 in senescence induction in response to irradiation.

## Discussion

Glioblastoma remains an incurable disease, with limited treatment options. Surgical resection, followed by radiation and chemotherapy remains the most effective treatment strategy. A better understanding of the mechanisms underlying the therapeutic response to radiation may help enhance treatment efficacy. Here we show that radiation response correlates with different molecular signatures in male and female human GBM lines. Using a mouse model of GBM, we further identify increased sensitivity to radiation in female GBM astrocytes, and determine that both non-apoptotic cell death and cellular senescence likely contribute to sex differences in radiation response. With a correlation-based approach, we identified p21 as a likely mediator of irradiation-induced senescence. Strikingly, the relationship between p21 and the senescence marker SA-β-gal significantly differed in male and female cells—a finding that was observed in mouse GBM model astrocytes, mouse wildtype astrocytes, and primary human GBM lines. Female cells had higher levels of senescence, as measured by SA-β-gal, in response to the same levels of p21—suggesting that p21 plays a more critical role in irradiation-induced cellular senescence in female than in male cells. This finding was confirmed by p21 knockdown, which decreased the percentage of senescent cells in female GBM model astrocytes to levels similar to those of males. Importantly, the relationship between SA-β-gal and p21 was evident across the heterogeneous p53 status of the varying murine models (wildtype, dominant negative, CRISPR deletion) and the heterogeneous nature of the human GBM cell lines. Together, these results indicate that the sex differences in p21-induced senescence are p53 independent. Finally, using a novel FCG model of GBM, which enables mechanistic dissection of the biological underpinnings of a sex difference, we determined that the relationship between p21 and irradiation-induced cellular senescence is patterned by gonadal sex. This finding will help guide future studies that aim to uncover the pathways regulating the p21-senescence axis and eventually modulate this interaction to improve radiation response in both sexes.

Our results highlight some intriguing directions for future studies. Interestingly, when p21 was knocked down in female GBM model astrocytes, it did not eliminate the senescence response to irradiation, but did abrogate the sex difference in

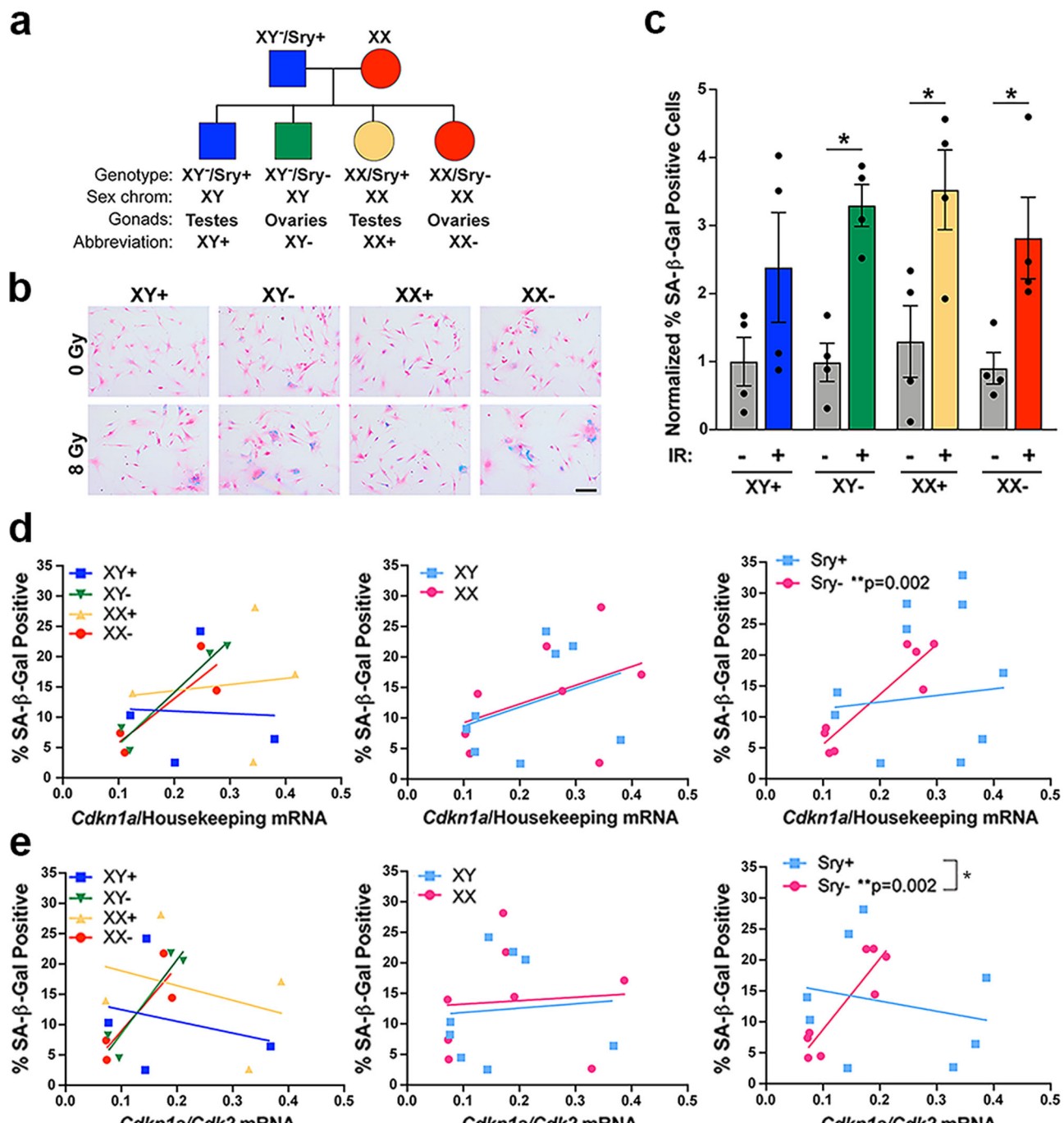

**Fig. 8 Gonadal sex patterns the relationship between p21 and SA-β-gal positivity. a** Diagram of the four genotypes resulting from the FCG mouse model. **b** Example images of FCG GBM model astrocytes stained for SA-β-gal 5 days after irradiation with 0 or 8 Gy, followed by counterstaining with nuclear fast red. Scale bar, 150 μm. **c** Quantification of the percentage of SA-β-gal positive cells. Results are from two cell lines (each consisting of a corresponding XY +, XY−, XX+, and XX− line generated at the same time) tested in two separate experiments. Values were normalized to the XY + 0 Gy condition, which was arbitrarily set at 1. *$p < 0.05$ as indicated by bracket. Data are means $+/-$ SEM ($n = 4$/genotype/condition). **d** Correlation between *Cdkn1a* (p21) mRNA levels at 24 h after irradiation with 0 or 8 Gy and the percentage of SA-β-gal positive cells at 5 days after irradiation in FCG GBM model astrocytes, showing each genotype separately (left), as well as grouped by sex chromosomes (center) or gonadal sex (Sry status) (right). Sry+ correlation: $r = 0.13$, $p = 0.7617$, Sry- correlation: $r = 0.91$, $p = 0.0016$ ($n = 8$/group). **e** Correlation between the *Cdkn1a/Cdk2* mRNA ratio at 24 h after irradiation with 0 or 8 Gy and the percentage of SA-β-gal positive cells at 5 days after irradiation in FCG GBM model astrocytes, showing each genotype separately (left), as well as grouped by sex chromosomes (center) or gonadal sex (Sry status) (right). Sry+ correlation: $r = -0.22$, $p = 0.5964$, Sry− correlation: $r = 0.91$, $p = 0.002$. Sry+ slope vs Sry− slope: $p = 0.0262$—indicated by bracket ($n = 8$/group).

response. This suggests that there are additional pathway(s), beyond p21, contributing to senescence induction after radiation, and that these are shared by males and females. Which signaling pathways, and whether they are always active or are upregulated with p21 loss, remains to be determined. Another avenue of future direction is uncovering how gonadal sex patterns the

relationship between p21 and senescence. Our findings suggest that the testosterone surge in utero (or lack thereof in females) is responsible for the sex differences in the p21-senescence axis. The hormone surge has been shown to exert long-term organizational effects in the brain through epigenetic mechanisms[1,31]. However, the differences we observe are not explained through simple regulation of accessibility at the p21 locus, since equivalent levels of p21 in males and females correspond to different percentages of SA-β-gal positive cells. Instead, differences in hormone exposure in utero may regulate proteins that influence p21 localization or activity through post-translational modifications[32–35]. Future studies will investigate this possibility, with the goal of enhancing the senescence response after irradiation.

We also observed a sex difference in the levels of SA-β-gal cells in male and female wild-type astrocytes with advanced population doublings in vitro. This suggests female cells may be more susceptible to replicative senescence and/or to oxidative stress, since cells were grown at supraphysiological oxygen levels[27]. This has important implications for the fields of aging and neurodegenerative disease. Senescent cells contribute to a wide range of normal and pathological changes with aging[36,37], and are increasingly thought to play a role in neurodegenerative disease[38,39]. Sex differences in p16 and p21 expression in aged mice in vivo have been reported, although this study actually found higher levels in male mice than female mice[40]. In our study, the same levels of p21 were associated with increased SA-β-gal cells in females, raising the possibility that this difference may not actually translate into increased senescent cells in males. Alternatively, it is possible that the senescent cell phenotype may also differ between male and female cells. The senescence-associated secretory phenotype varies based on cell type and senescence induction mechanism[21,41], and so could reasonably be influenced by cell sex. This could potentially result in different rates of clearance for male and female senescent cells in vivo, or mean that the same levels of senescence have different effects on the surrounding tissue depending on sex.

One of the potential limitations of our study is that we rely on SA-β-gal for our primary measure of senescence. While SA-β-gal is the most widely used marker of senescence, there is debate about its specificity, since it is not required for senescence induction and has been observed in non-senescent cells[19]. In support of the idea that SA-β-gal does correspond to senescence in our study: (1) it increased with exposure to radiation in a dose-dependent manner, (2) along with increases in SA-β-gal, we saw more cells with an enlarged, irregular shape, (3) cell cycle arrest was confirmed by a decrease in EdU incorporation and in the percent of cells positive for the proliferation marker Ki67, (4) expression changes consistent with ionizing radiation-induced senescence were observed in our cells, and (5) levels of the cell cycle inhibitor p21 also increased. Regardless of whether SA-β-gal is a measure of true senescence, our findings still have important implications for the field. When interpreting studies that use p21 as a marker of senescence, it may be necessary to consider the sex of the subjects, and whether the same pathway could have differences in downstream outcomes between male and female cells.

Finally, our results highlight the utility of correlation analyses when studying sex differences. Most phenotypes are not truly sexually dimorphic, but represent a spectrum of values in males and females. While the mean values differ, and the two ends of the spectrum are populated primarily by individuals of one sex or another, there is considerable overlap. In addition, the variability between individuals of a single sex may obscure the differences between the two sexes, requiring a large sample size to detect the difference. With our approach, we discovered a difference in the relationship between p21 and SA-β-gal in males and females that was not apparent when we simply compared mean p21 levels alone. This could offer a powerful investigation strategy that takes advantage of individual variability rather than viewing it as a liability.

In summary, we have uncovered a sex difference in the relationship between p21 and irradiation-induced cellular senescence that has potential implications for the fields of cancer research, neuroscience, and aging. Better understanding of this mechanism could identify novel approaches to improve response to cancer therapy.

## Methods

**Primary human GBM lines**. Primary human GBM lines were kindly provided by Dr. Albert H. Kim. Specimens for culture were obtained prospectively at the time of surgery and cell lines were established[42]. Consent was obtained in accordance with a Washington University Institutional Review Board (IRB) approved Human Studies Protocol. Patient charts were reviewed for the molecular characteristics of the tumors/GBM lines (Supplementary Table 1). Available information varied between tumors; where no information was available, that cell in the table was left blank. Human GBM lines were grown on laminin (Sigma) coated Primaria plates in RHB-A media (Takara) supplemented with growth factors EGF (Sigma) and bFGF (Millipore) at 50 ng/ml. Media was replaced with half fresh media every 2–3 days. For passaging, cells were detached with Accutase (Sigma) and split 1:2 to 1:4 depending on cell line. One female human line (B51) was excluded from the SA-β-gal studies, since these cells did not attach to glass coverslips, even when coverslips were coated with laminin.

**Mouse wild-type astrocytes**. All animals were used in accordance with an Animal Studies Protocol approved by the Animal Studies Committee of the Washington University School of Medicine, per the recommendations of the Guide for the Care and Use of Laboratory Animals (NIH). Mouse wildtype astrocytes were isolated from the cortex of postnatal day 1 C57BL/6J pups as described[13]. Briefly, the cortices were dissected from the rest of the brain in cold HBSS and the meninges were removed. Isolated cortices were incubated in 500 μl 0.05% Trypsin-EDTA + 0.02 mg/mL DNase I (Roche) for 15 min at 37 °C. Trypsin was neutralized by the addition of 750 μl media (DMEM/F12 + 10% FBS + 1% penicillin-streptomycin), and tissue was triturated 10–15 times with a p1000 pipet, then filtered through a 40 μm cell strainer. Cell suspension was spun down, and the pellet was resuspended and plated in a poly-L-lysine (ScienCell) coated T25 flask. Cells were grown until confluent (~7 days). To obtain a pure population of astrocytes, confluent flasks were shaken at 225 rpm overnight at 37 °C. After shaking, the media was aspirated, removing any floating cells, and flasks were rinsed once with sterile PBS. The remaining attached cells, representing the astrocyte population, were trypsinized and replated in Primaria T75 flasks (Corning). Cells were then expanded for downstream analyses. Prior to experimental treatment, cells were switched to media without penicillin/streptomycin (DMEM/F12 + 10% FBS). For the irradiation studies, all data points represent astrocyte cultures from individual pups. The sex of each culture was determined by genotyping the corresponding pup tail DNA for the X and Y chromosome paralogs Kdm5c (Jarid1c) and Kdm5d (Jarid1d) (Forward: CTG AAG CTT TTG GCT TTG AG, Reverse: CCA CTG CCA AAT TCT TTG G)[43]. All irradiation studies were carried out on astrocytes that were passage 2 or 3. For the studies of astrocytes at low and high population doubling levels, cultures from multiple pups were pooled by sex to generate male and female cultures.

**Mouse GBM model astrocytes**. Briefly, astrocytes were isolated from the cortex of postnatal day 1 Nf1$^{flox/flox}$;GFAP-Cre mouse pups as detailed above. Sex was determined by genotyping pup tail DNA for the X and Y chromosome paralogs Kdm5c (Jarid1c) and Kdm5d (Jarid1d)[43]. Astrocytes from at least three male and three female pups were then pooled by sex. Male and female Nf1−/− astrocyte cultures were infected with a retrovirus encoding EGFP and a flag-tagged dominant-negative p53 (DNp53), consisting of amino acids 1–14 of the transactivation domain, followed by amino acids 309–393. GFP positive-DNp53 expressing cells were selected for using fluorescence-activated cell sorting (FACS). Nf1−/− DNP53 cells were cultured in DMEM/F12 supplemented with 10% FBS and 1% penicillin-streptomycin. Five individually generated Nf1−/− DNp53 cell lines, each consisting of a paired male and female line, were used for the experiments in this study[13].

**p21 knockdown lines**. Briefly, male and female Nf1−/− DNP53 astrocytes were infected with the lentiviral Cas9/sgRNA-p21 all-in-one construct (p21 sgRNA sequence: ACTTCGTCTGGGAGCGCGTT). Cas9 control lines were generated by infecting with the lentiviral Cas9 vector without any guide RNA. Following infection, cell lines were selected by puromycin treatment (2.5 μg/ml) for 1–2 weeks. Surviving cells were expanded and p21 knockdown was confirmed by western blot[14].

**FCG GBM model astrocytes**. To generate the FCG-GBM model, Four Core Genotypes XY-/Sry+ male mice[30,44–46] (Jackson Laboratory #010905) were bred with constitutive Cas9 expressing[47] female mice (Jackson Laboratory #026179), and astrocytes were isolated from the cortex of postnatal day 1 FCG-Cas9 mouse pups as described[13]. Pup tail DNA was genotyped for Sry and the Y chromosome following the protocol available from Jackson Laboratory (Sry Forward: AGC CCT ACA GCC ACA TGA TA, Sry Reverse: GTC TTG CCT GTA TGT GAT GG, Y Chrom. Forward: CTG GAG CTC TAC AGT GAT GA, Y Chrom. Reverse: CAG TTA CCA ATC AAC ACA TCA C, Internal Control Forward: CAA ATG TTG CTT GTC TGG TG, Internal Control Reverse: GTC AGT CGA GTG CAC AGT TT). Astrocytes from at least three pups from each of the four genotypes were pooled to generate XY⁻/Sry+ (XY+), XY⁻/Sry− (XY−), XX/Sry+ (XX+), and XX/Sry− (XX−) cultures. FCG-Cas9 astrocytes at passage 4 were transfected with a modified pX330 vector with dual sgRNAs targeting Nf1 and p53 (Nf1 sgRNA sequence: GCAGATGAGCCGCCACATCGA, p53 sgRNA sequence: CCTCGAGCTCCCTCTGAGCC). The pX330-Nf1-p53 vector was kindly provided by Dr. Kwanha Yu and Dr. Benjamin Deneen (Baylor College of Medicine). FCG-Cas9 astrocytes that received the pX330-Nf1-p53 plasmid were immortalized and had a significant growth advantage. Since wildtype astrocytes stop dividing after passage 5–6, growth advantage was used to select for astrocytes with successful CRISPR mutation/deletion of Nf1 and p53. FCG-Cas9 Nf1/p53 CRISPR astrocytes were passaged at least three times, then protein was collected for confirmation of Nf1 and p53 knockdown by western blot.

**Irradiation**. For irradiation experiments, cells were irradiated using an RS 2000 X-ray irradiator (Rad Source Technologies) unless otherwise specified in the specific methods section. Radiation was delivered at a dose rate of ~1.8 Gy/min with 160 kVp X-rays. Control cells were transported to the irradiator and sat on the bench for the same length of time as cells were in the irradiator.

**Human GBM irradiation dose-response curves**. Human GBM cells were lifted with Accutase, counted, and 20,000 cells per well were plated in laminin (Sigma) coated 24-well plates. Cells were plated in triplicate wells for each irradiation dose. Cells were irradiated 24 h later with 0, 2, 4, 6, 8, or 10 Gy, then allowed to grow for 4 days, with fresh media added to the wells 2 days after irradiation. On day 4, media was aspirated off, and 200 μl of Accutase was added to each well. Plates were incubated for 3 min in a 37 °C incubator, then wells were rinsed with Accutase to detach all cells, and cell suspension was transferred to 1.7 ml tubes on ice. Cell suspension was diluted 1:1 with Trypan blue, and cells were counted with a hemocytometer. Three technical replicates were counted per dose. Percentage of cell number was calculated by dividing the counts for each dose by the average cell number for the 0 Gy condition. For all cell lines except B5, two separate dose curves were performed, and the six technical replicates were combined to get the final irradiation dose response curve. Due to slow growth rate, B5 cell number was limited, and only one dose curve, with three technical replicates, was performed.

**Mouse GBM irradiation dose response curves**. Irradiation dose response curves for mouse GBM model astrocytes were performed using the sulforhodamine B assay as described, with minor modifications[48]. Briefly, Nf1−/− DNp53 astrocytes were trypsinized, counted, and 1000 cells per well in 200 µl media were plated in 96-well white-walled clear-bottom plates (Corning). Six technical replicate wells were plated for each irradiation dose. A standard curve for each cell line and sex was plated at the same time by performing a twofold serial dilution starting at 100,000 cells per well, and diluting six times, plating 0 cells in the final row. Standard curve wells were plated in triplicate for each cell line. 6 h after plating, standard curves were fixed by the addition of 100 µl ice cold 5% trichloroacetic acid and incubating at 4 °C for 1 h. Standard curve plates were then washed four times with tap water and allowed to dry overnight. 24 h after plating, cells were irradiated with 0, 2, 3, 4, 6, 8, or 9 Gy, then allowed to grow for 4 days. On day 4, plates were fixed with 5% trichloroacetic acid, then washed and dried. After drying, plates were stained by adding 100 µl of 0.057% sulforhodamine B (SRB) diluted in 1% acetic acid to each well and incubating for 30 min at room temperature. Plates were then washed three times with 1% acetic acid and allowed to dry overnight. SRB was solubilized by adding 100 µl 10 mM Tris-Base (pH 10.5) to each well, then incubating for 1–2 h at room temperature on a shaker. Absorbance was measured at 510 nm using an Infinite 200 PRO microplate reader (Tecan). Standard curve absorbance values were graphed using GraphPad Prism software, and a linear standard curve was fit. The standard curve for each cell line was used to interpolate the cell number for experimental wells. Percentage of cell number was calculated by dividing the cell counts for each irradiated well by the average cell number for the 0 Gy condition. Technical replicates were averaged to derive a single value for each dose per cell line and sex, and results from each of the 5 Nf1−/− DNp53 cell lines were combined to generate the final irradiation dose response curves for male and female mouse GBM model astrocytes.

**Cell growth assays via live cell imaging**. Cells were plated at 1000 cells per well into a 96-well plate with five technical replicates plated per condition. Cells were irradiated using a GammaCell 40 Irradiator (Best Theratronics). Non-irradiated plates were transported to the radiation facility, but not exposed. The 96-well plates

were then immediately placed into the IncuCyte ZOOM live cell imaging system (Satorius). Phase-contrast images were taken every 4 h for a total of 72 h with 4 scan areas taken per well. Cell confluence was analyzed using the IncuCyte ZOOM analysis software. Percent confluence over time was used as a measure of longitudinal cell growth.

**Clonogenic assay**. Cells were plated at 500 cells per well into six-well plates. The next day, cells were either left unirradiated (0 Gy) or irradiated with 4, 8, 12, or 16 Gy using a GammaCell 40 Irradiator (Best Theratronics). Media was changed 24 h after irradiation. Five days post-treatment, cells were washed in 1× PBS, before being fixed with 100% methanol for 10 min and stained with a 0.5% crystal violet solution for 30 min. Plates were imaged the next day on the LI-COR Odyssey near infrared imaging system and analyzed via a custom ImageJ macro which counts individual colonies, allowing for unbiased quantification.

**Immunofluorescence for γH2AX**. Cells were plated at 20,000 cells per well onto coverslips within the wells of 24-well plates. Twenty-four hours post plating, cells were left un-irradiated (0 Gy) or irradiated with 3 Gy or 8 Gy using a GammaCell 40 Irradiator (Best Theratronics). Coverslips were harvested and fixed with MeOH at 1, 6, 24, and 48 h post-irradiation. Cells were then processed for immuno-fluorescence for γH2AX (1:500; Millipore, 05–636 JBW301). Secondary detection was accomplished using Alexa Fluor 488 goat anti-mouse IgG (1:500; Invitrogen, A28181). Nuclei were counterstained with Hoechst. Images were acquired using a Leica DM5500B upright epifluorescence microscope for a total number of 100–200 cells analyzed per n. Cells were scored as having more or less than 10 γH2AX foci.

**Western blot analysis**. For protein collection, plates were rinsed once with cold PBS, then cells were lysed with RIPA buffer plus cOmplete™ Protease Inhibitor Cocktail (Roche), PhosStop Phosphatase Inhibitor Cocktail (Roche), and PMSF. Cell lysates were left on ice for 10 min, vortexing every 2 min, then frozen at −80 °C. Before use, lysates were thawed and spun at 16,000 × g for 15 min at 4 °C. Supernatant was collected and used for downstream analyses. Protein concentration was measured with the DC Protein Assay (Bio-Rad), following the manufacturer's microplate assay protocol, and using an Infinite 200 PRO microplate reader (Tecan) to measure absorbance. Total protein lysate was combined with NuPAGE LDS Sample Buffer (Invitrogen) and NuPAGE Sample Reducing Agent (Invitrogen), then samples were heated for 10 min at 95 °C. For Nf1−/− DNp53 astrocyte and Nf1−/− DNp53 Cas9 and p21 KD samples, 100 μg of protein was loaded, and samples were run on a NuPAGE 4–12% Bis-Tris Gel in MES SDS Running Buffer (Invitrogen). For wild-type mouse astrocyte and human GBM samples, 30 μg of protein was loaded, and samples were run on a NuPAGE 4–12% Bis-Tris Gel in MES SDS Running Buffer (Invitrogen). For FCG-Cas9 and FCG-Cas9 Nf1/p53 CRISPR samples, 75 μg of protein was loaded and samples were run on a NuPAGE 4–12% Bis-Tris Gel in MOPS SDS Running Buffer (Invitrogen). After separation by electrophoresis, proteins were transferred to Odyssey nitro-cellulose membrane (LI-COR). Membranes were blocked for 1 h at room temperature in Odyssey Blocking Buffer (LI-COR) or Intercept (PBS) Blocking Buffer (LI-COR) diluted 1:1 with PBS-T. Membranes were incubated overnight in primary antibody diluted in blocking buffer (1:1000 anti-cleaved caspase-3 Cell Signaling Technologies #9664, 1:4000 anti-α-Tubulin Sigma #T5168, 1:1000 anti-PARP Cell Signaling Technologies #9542, 1:2000 anti-mouse p21 Abcam #ab188224, 1:1000 anti-Cdk2 Cell Signaling Technologies #2546, 1:1000 anti-human p21 Cell Signaling Technologies #2947, 1:200 anti-Nf1 Santa Cruz #sc-376886, 1:1000 anti-p53 Sigma #T5168, 1:25,000 anti-Actin Sigma #A1978). Membranes were washed with PBS-T then incubated in secondary antibody diluted in blocking buffer for 1 h at room temperature (1:30,000 IRDye 680RD Donkey anti-Rabbit or Donkey anti-mouse, 1:20,000 IRDye 800CW Donkey anti-Rabbit or Donkey anti-mouse (LI-COR)). Membranes were washed in PBS-T, then proteins were visualized using a ChemiDoc MP Imaging System (Bio-Rad). Band intensity was quantified using Image Lab Software Version 6.0.0 (Bio-Rad).

**Flow cytometric analysis of EdU/PI and Annexin-V staining**. Cells were seeded at a density of $2 \times 10^5$ (females) and $1 \times 10^5$ (males) in T25 flasks and cultured in DMEM/F12 with 10% FBS and 1% penicillin-streptomycin. The next day cells were treated with 0, 3, 6 or 8 Gy irradiation (IR) and returned to the incubator. Cell cycle analysis was conducted at 24 h after irradiation. Annexin-V staining was conducted at 24 h and 5 days after irradiation. Briefly, cells were treated with 10 μM EdU for 1 h in their optimum growth culture condition to allow for EdU incorporation into active DNA synthesis. They were then harvested using trypsin, along with their media to include the floaters, and counted using a Countess 3 automated cell counter (Invitrogen, AMQAX2000). Live versus dead cells were determined using trypan blue exclusion. Cells were strained through a 70 µM filter and divided into two tubes, one for Annexin-V labeling, and one for EdU-PI labeling. A total of $3 \times 10^5$ cells were used for each stain. Annexin-V: For annexin-V staining, cells were labeled with 5 µl Annexin-V (Pacific Blue, Invitrogen, A35122) in 100 μL 1X Annexin-V binding buffer (Invitrogen, V13246) for 15 min in the dark at room temperature. After 15 min incubation, 400 µl of 1X Annexin-V binding buffer was added to each sample and FACS analyzed using the CytoFlex S Flow Cytometer (Beckman Coulter, CO2949). Etoposide (final concentration of 20 µg) was used as a

positive indicator of apoptosis. Edu-PI: For EdU-PI staining, cells were prepared and resuspended in 1X Click-iT™ permeabilization and wash reagent per the manufacturer's instructions (Invitrogen, C10636). Detection of Click-iT EdU was followed immediately by incubating the cells with the Click-iT Plus reaction cocktail (438 μL PBS, 10 μL copper protectant, 2.5 μL fluorescent dye picolyl azide, 50 μL reaction buffer additive per sample) for 30 min at room temperature, protected from light. Cells were then washed with 3 mL 1X Click-iT™ permeabilization and wash reagent, centrifuged, and resuspended in 200 μL of 1X Click-iT permeabilization and wash reagent containing 0.3 μL PI, and FACS analyzed using the CytoFlex S Flow Cytometer (Beckman Coulter, CO2949). Unstained samples and cells that had been serum-starved for 48 h were used as controls for demarcating background/autofluorescence staining and to demarcate the cell cycle arrest at G0/G1, respectively.

**Senescence-associated β-galactosidase staining.** For *Nf1−/− DNp53* astrocytes and FCG GBM model astrocytes, cells were plated in six-well plates at ~70% confluence (70,000–100,000 cells per well depending on cell line/sex) and irradiated 24 h later. Four days after irradiation, cells were lifted, counted, and plated on poly-L-lysine (ScienCell) coated glass coverslips in 24-well plates at sub-confluency (15,000 cells per well). Twenty-four hours later, cells were stained using the Senescence Associated β-galactosidase Staining Kit from Cell Signaling Technologies, according to the manufacturer's instructions. Briefly, cells were washed once with PBS, then fixed for 15 min with provided Fixative Solution. After fixing, cells were rinsed twice with PBS, then incubated in β-galactosidase staining solution (pH 5.9–6.1) overnight (~16 h) in a 37 °C dry incubator. The next morning, staining solution was removed and cells were rinsed twice with PBS. For mouse wild-type astrocytes, cells were grown in Primaria 6 well plates and irradiated at ~70% confluence. 5 days after irradiation, cells were lifted, counted, and plated on poly-L-lysine coated glass coverslips in 24-well plates at sub-confluency (27,500 cells per well). Forty-eight hours later, cells were stained using the Senescence Associated β-galactosidase Staining Kit. For human GBM lines, cells were grown in laminin-coated six-well plates and irradiated at ~70% confluence. Four days after irradiation, cells were lifted, counted, and plated on laminin-coated glass coverslips in 24-well plates at sub-confluency (25,000 cells per well). Twenty-four hours later, cells were stained using the Senescence Associated β-galactosidase Staining Kit. After SA-β-gal staining, cells were counterstained with Nuclear Fast Red (Vector Laboratories), following the manufacturer's instructions. Coverslips were dehydrated then mounted on slides using Permount Mounting Medium (Fisher) diluted 2:1 with Xylene substitute. Images were taken on a Zeiss Axio Scope A1 upright light microscope at 10× magnification. Total cells and positive cells were counted by hand in ImageJ (FIJI Version 2.1.0) using the Cell Counter plugin. All images were taken and quantified by an experimenter blinded to the sex and experimental condition of the sample.

**Immunocytochemistry for Ki67 and Geminin.** Ki67: *Nf1−/− DNp53* astrocytes were plated in six-well plates at ~70% confluence and irradiated the following day. Four days after irradiation, cells were trypsinized and counted, and 15,000 cells per well were plated on poly-L-lysine (ScienCell) coated glass coverslips in 24-well plates. Twenty-four hours later cells were washed once with PBS, then fixed with 3.2% paraformaldehyde (Electron Microscopy Sciences) for 15 min. Cells were then washed twice with PBS and stored at 4 °C until staining was performed. For Ki67 immunocytochemistry, cells were permeabilized with 0.2% Triton-X 100 for 15 min at room temperature and nonspecific antibody staining was blocked with antibody diluent (10% normal donkey serum, 1% bovine serum albumin (BSA) and 0.2% Triton-X 100 in PBS) for 1 h at room temperature. Cells were then incubated in rabbit anti-Ki67 (Abcam ab15580, 1:500) or rabbit IgG control (Cell Signaling) overnight at 4 °C, after which cells were washed and incubated in donkey anti-rabbit Alexa Fluor 555 (Invitrogen, 1:1000) for 1 h at room temperature. Nuclei were stained with DAPI for 1 min, then coverslips were mounted in Immu-Mount (Thermo Fisher Scientific). Images were taken on an Olympus BX60 (Olympus, Japan) fluorescence microscope at 10× magnification. Ki67 images were taken at 500 ms exposure. Color images were converted to 16 bit and thresholded in ImageJ. Thresholds were determined using IgG controls. After thresholding, cell number was counted with the automated cell counter (Analyze Particles); anything <60 pixel$^2$ was excluded from the count. The positive rate is the ratio of stained cells to total cells. All images were taken and quantified by an experimenter blinded to the sex and experimental condition of the sample. Geminin: Mouse wildtype astrocytes were grown in Primaria 6 well plates and irradiated at ~70% confluence. Five days after irradiation, cells were lifted, counted, and plated on poly-L-lysine coated glass coverslips in 24-well plates at sub-confluency (27,500 cells per well). Forty-eight hours later, cells were washed once with PBS, then fixed with 3.2% paraformaldehyde for 15 min. Cells were then washed twice with PBS and stored at 4 °C until staining was performed. For Geminin staining, cells were permeabilized with 0.2% Triton-X 100 in PBS, and processed for immunofluorescence as described above for Ki67. Geminin was localized with anti-geminin antibody (Abcam ab175799, 1:1000) and secondary detection was accomplished with donkey anti-rabbit Alexa Fluor 555 (Invitrogen a31572, 1:1250). Total numbers of geminin-positive cells were counted by a blinded observer.

**RNA isolation and cDNA preparation – Mouse GBM model astrocytes.** RNA was isolated from *Nf1−/− DNp53* astrocytes and FCG GBM model astrocytes using TRIzol Reagent (Invitrogen), following the manufacturer's protocol. Briefly, cells were grown in 10-cm dishes to ~70% confluence, then irradiated. At desired timepoint after irradiation, media was aspirated, and cells were washed once with cold PBS. 1 ml of Trizol was added to the dish; cells were scraped into Trizol, allowed to sit 5 min at room temperature, then transferred to a 1.7 ml tube. 200 μl of chloroform was added, and samples were shaken vigorously for 15 s, then sat at room temperature for 3 min. Samples were spun at $12,000 \times g$ for 15 min at 4 °C. The aqueous phase was transferred to a new tube, and 500 μl of isopropyl alcohol was added to precipitate RNA. Samples were incubated for 10 min at room temperature, then spun at $12,000 \times g$ for 10 min at 4 °C. Supernatant was aspirated, and pellet was washed with 1 ml cold 75% ethanol. Samples were spun at $7500 \times g$ for 5 min at 4 °C and ethanol was aspirated. After drying, pellet was resuspended in 35 μl molecular biology grade water (DNase/RNase free). RNA concentration was measured with a NanoDrop 1000 spectrophotometer (Thermo Scientific). RNA was treated with Amplification Grade DNAse I (Invitrogen) to eliminate genomic DNA, and cDNA was generated using the SuperScript III First-Strand Synthesis System (Invitrogen), according to the manufacturer's instructions.

**RNA isolation and cDNA preparation – Mouse wildtype astrocytes.** RNA was isolated from mouse wildtype astrocytes using the QIAGEN RNeasy Mini Kit, according to the manufacturer's instructions. Briefly, astrocytes were grown in 10-cm Primaria dishes to ~70% confluence, then irradiated. At desired timepoint after irradiation, media was aspirated, and cells were washed once with cold PBS. 350 μl of RLT buffer + β-mercaptoethanol was added and cells were scraped into RLT buffer, then transferred to a 1.7 ml tube. Cells were homogenized by passing through a 20-gauge needle attached to a 1 ml syringe ~10 times, then 350 μl of 70% ethanol was added to the cell lysate and mixed by pipetting, before transferring to the RNeasy spin column. To eliminate genomic DNA, the optional on column DNase digestion was performed by adding 80 μl DNase I solution (QIAGEN) to the column membrane and incubating 15 min at room temperature. RNA was eluted by adding 30 μl molecular biology grade water (DNase/RNase free) directly to the column membrane, incubating at room temperature for 2.5 min, then spinning at $10,000 \times g$ for 1 min. RNA concentration was measured with a NanoDrop 1000 spectrophotometer (Thermo Scientific). cDNA was generated using the QuantiTect Reverse Transcription Kit (QIAGEN) following the manufacturer's instructions.

**Quantitative real-time PCR.** Quantitative RT-PCR was performed using gene-specific primers and iTaq Universal SYBR Green Supermix (Bio-Rad). A standard curve was generated for each experiment by pooling cDNA from all samples and serially diluting to generate concentrations of 25, 12.5, 6.25, and 3.125 ng/μl. Samples were diluted to a concentration of 10 ng/μl. 2 μl of sample or standard was loaded in each well, and two technical replicates were run for each sample. A CFX Connect Real-Time PCR Detection System (Bio-Rad) was used to collect and analyze the amplification data. Analyses of standard curves, melt curves, and reverse-transcriptase-negative controls were used to verify primer and amplification reaction quality. Expression levels of genes of interest were normalized to the average expression of three housekeeping genes: *Gapdh*, *Actb*, *Rpl5*. Primer sequences: *Actb* (Forward: TGT ATT CCC CTC CAT CGT G, Reverse: CGC AGC TCA TTG TAG AA GG), *Ccnd1* (Forward: GTG CAT CTA CAC TGA CAA CTC, Reverse: TGG TCT GCT TGT TCT CAT CC), *Cdk2* (Forward: GCA CCA GGA CCT CAA GAA AT, Reverse: ACG GTG AGA ATG GCA GAA AG), *Cdkn1a* (Forward: AGA CAT TCA GAG CCA CAG GCA CCA, Reverse: GCA TCG CAA TCA CGG CGC AA), *Cdkn1b* (Forward: TGG TGG ACC AAA TGC CTG AC, Reverse: TTC GGG GAA CCG TCT GAA AC), *Cdkn2a* (Forward: GAC ATC GTG CGA TAT TTG CGT TCC G, Reverse: TTT AGC TCT GCT CTT GGG ATT GGC C), *Gapdh* (Forward: GGC AAA TTC AAC GGC ACA GT, Reverse: AGA TGG TGA TGG GCT TCC C), *Rpl5* (Forward: GGA AGC ACA TCA TGG GTC AGA, Reverse: TAC GCA TCT TCA TCT TCC TCC ATT), *Xist* (Forward: GCT GTA GTA GTC ACA GTC CCA, Reverse: CTG TGT TTG CCC CTT TGC TA).

**Statistics and reproducibility.** Data were graphed and statistical analyses, including two-way ANOVAs, t-tests, one sample t-tests and dose response curves, were performed using GraphPad Prism software (Prism 9 for macOS Version 9.0.0). All tests were two-tailed unless otherwise specified. The two-way ANOVA models fit the effect of sex, time or dosage, and their interaction. Dose response curves were fit to normalized response versus dosage using the 4-parameter logistic regression dose-response model with the slopes estimated by maximizing the likelihood function. For Nf1−/− DNp53 graphs normalized to the Male 0 Gy condition of the same cell line (Figs. 3a–c, 7b–d, and Supplementary Fig. 2), the Male 0 Gy values were derived by dividing by the Male 0 Gy average. Spearman rank-based correlation analysis between IC50 and gene expression was performed as described[7] using R (version 3.5.0). Briefly, Spearman correlation coefficients of irradiation IC50 values with expression of either MC5 genes (17 genes), FC3 genes (9 genes), or randomly selected gene sets of the same size, were calculated. The Olkin-averaged Spearman correlation coefficients with IC50 across MC5 genes and across FC3 genes per cell line was summarized and tested against 0 with those

derived for 1000 random gene sets serving as negative control. To explore the relationship between senescence and the various molecular markers focused on p16, p21, and p21/CDK2 ratio, linear regression analysis and Pearson's correlation analysis was conducted using R (version 4.0.2). Pearson's correlation coefficients (r) were calculated and the Fisher's Z transformed coefficients were compared between sex by normal test. In the integrative linear regression modeling across sex, the dependent/outcome variable was senescence, as measured by SA-β-gal, while the independent/predictor variable included the continuous mRNA or protein levels of the molecular marker of interest and sex. To assess the possible linear association between SA-β-gal and each of the continuous variables by sex, sex-specific intercepts and slopes were estimated from the integrative linear model. The slope difference between the male and female-specific slopes was derived with 95% confidence interval (CI) and tested against 0 by the two-sided Wald test. Significance was claimed at $p < 0.05$. Detailed slope and correlation estimation results can be found in Supplementary Table 3. For the Nf1−/− DNp53 mRNA correlations only, mRNA expression levels from cells irradiated with 0, 6 or 9 Gy were correlated with SA-β-gal results from cells irradiated with 0, 6, or 8 Gy—thus resulting in a correlation consisting of 0, 6, and 8/9 Gy doses. For all other correlations, irradiation levels were exactly the same for protein/mRNA measures and SA-β-gal.

**Reporting summary**. Further information on research design is available in the Nature Research Reporting Summary linked to this article.

## Data availability

The source data for all figures and tables is presented in the Supplementary Data file. Uncropped and unedited western blot images are also available in the Supplementary Data File. Any additional details can be requested from the corresponding author.

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

## Acknowledgements

The authors would like to thank the Department of Radiation Oncology, Washington University School of Medicine for use of the RS 2000 X-ray irradiator. They also thank William Eades and the Siteman Flow Cytometry Core at Washington University School of Medicine for their help with flow cytometry. The Siteman Cancer Center Flow Cytometry Shared Facility is supported in part by an NCI Cancer Center Support Grant P30 CA091842. This work was supported by NIH R01 CA174737-06 (J.B.R.), NIH R21 NS098210 (J.B.R.), Joshua's Great Things (J.B.R.), NIH R01 NS094670 (A.H.K.), NIH R01 NS106612 (A.H.K.), Siteman Investment Program Pre-R01 Research Development Award (A.H.K.), NIH R03 CA227206 (M.V.), a Research Scholar Grant RSG-18-066-01-TBG from the American Cancer Society, funds from The Ohio State University Comprehensive Cancer Center/Department of Radiation Oncology (M.V.), and the National Institute of General Medical Sciences of the National Institutes of Health under award number T32GM068412-11A1 (M.M.T.). L.G. and C.M.H. were supported by the MARC U-STAR Program at Washington University in St. Louis, grant number T34 GM083914. J.L. also received support from the NCI Cancer Center Support Grant for the Siteman Cancer Center P30 CA091842 (PI: Dr. Timothy J. Eberlein).

## Author contributions

L.B.—Conception and design of the work; acquisition, analysis, and interpretation of data; figure preparation, manuscript draft and editing, approved the submitted version. N.M.W.—Design of the work; acquisition, analysis, and interpretation of data; figure preparation, manuscript editing, approved the submitted version. L.G.—Acquisition, analysis, and interpretation of data, manuscript editing, approved the submitted version. T.A.-A.—Design of the work; acquisition, analysis, and interpretation of data; figure preparation, manuscript editing, approved the submitted version. O.T.—Acquisition, analysis, and interpretation of data; figure preparation, manuscript editing, approved the submitted version. S.S.—Statistical analysis and interpretation of data; figure preparation, manuscript editing, approved the submitted version. M.M.T.—Acquisition, analysis, and interpretation of data; figure preparation, manuscript editing, approved the submitted version. G.R.—Acquisition, analysis, and interpretation of data; manuscript editing, approved the submitted version. W.Y.—Statistical analysis and interpretation of data; figure preparation, manuscript editing, approved the submitted version. J.S.—Acquisition, analysis, and interpretation of data; manuscript editing, approved the submitted version. L.Y.—Acquisition, analysis, and interpretation of data; figure preparation, manuscript editing, approved the submitted version. N.K.-B.—Reagent generation; analysis, and interpretation of data; manuscript editing, approved the submitted version. C.M.H.—Acquisition, analysis, and interpretation of data; manuscript editing, approved the submitted version. S.A.Q.—Acquisition, analysis, and interpretation of data; manuscript editing, approved the submitted version. D.D.M.—Reagent generation; manuscript editing, approved the submitted version. A.H.K.—Reagent generation; manuscript editing, approved the submitted version. S.A.S.—Design of the work; interpretation of data; manuscript editing, approved the submitted version. M.V.—Design of the work; acquisition, analysis, and interpretation of data; figure preparation, manuscript editing, approved the submitted version. J.L.— Design of the work; analysis and interpretation of data; figure preparation, manuscript editing, approved the submitted version. J.B.R.— Conception and design of the work; acquisition, analysis, and interpretation of data; figure preparation, manuscript draft and editing, approved the submitted version.

## Competing interests

The authors declare no competing interests.
