## [Peer Review file · Communications Biology]

Reviewers' comments:

Reviewer #1 (Remarks to the Author):

The present work focuses at sex-specific differences in the radiotherapy response of glioblastoma using murine and patient-derived cells as primary model systems. The authors identify senescence as major response mechanism and provide evidence for key biological differences between male and female cells. Precisely, they identify female astrocytes and transformed glioblastoma cells as being more sensitive to radiotherapy, which is mediated by higher percentages of p21-induced senescent cells. Ultimately, they leverage an innovative Four Core Genotypes mouse model of glioblastoma to identify hormonal effects in utero as causative factor.

Building on extensive previous research, the authors present novel work that is highly relevant and interesting to the community. The methods they apply are cutting-edge and their conclusions sound. The manuscript is elegant, fluent and highly pleasant-to-read.

I have two points.

1. The start comes somewhat abrupt. I understand that the authors build on gene signatures and clusters, which they have previously identified in human m/f tumor samples and confirmed in chemotherapy-treated m/f cells. Still, it is a very supervised approach with moderate, not too intuitive results that do not fully connect to previous findings. Are there independent radiotherapy-induced m/f-specific gene signatures?

2. The authors emphasize the relevance of using correlations as a statistical means to identify relevant biological associations. However, when looking at several scatter plots, e.g. in Figure 6, some correlations are driven by outliers and the exclusion of single data points would result in different results. I do believe that their findings are true since the same associations are found in non-neoplastic and neoplastic murine and human cells but it should be addressed as a potential limitation.

Reviewer #2 (Remarks to the Author):

This is an excellent study from an outstanding group who are leaders in the sex differentiation stratification of GBM patient outcomes. The experiments are well-controlled and the data are interpreted wisely.

There is only one comment to address here. Does any of this matter? It's already known that female patients get GBM less frequently than men, but in the current study, the authors primarily evaluate senescence parameters in response to irradiation (IR). In vivo analysis of whether there are differential growth patterns after IR with female vs. male GBM may allay this concern - or it could corroborate it. Either way, in vivo analysis of the currently in vitro study needs to be performed to support the in vitro outcomes that are in the current report.

Reviewer #3 (Remarks to the Author):

Broestl et al presented a paper entitled: Gonadal sex patterns p21-induced cellular senescence in mouse and human glioblastoma.

The authors state that males exhibit higher incidence and worse prognosis for the majority of cancers, including glioblastoma (GBM), and using a mouse model of GBM, they showed that female cells are more sensitive to radiation, and that both male and female GBM cells undergo radiation-induced p21-

dependent senescence. However, female cells were more sensitive to changes in p21 levels, as observed in both wild-type and transformed murine astrocytes and patient-derived GBM cell lines. Using a novel Four Core Genotypes model of GBM, we further show that sex differences in p21-induced senescence are patterned by gonadal sex. Collectively, their data suggested that sex differences in p21-induced senescence contribute to the female survival advantage in GBM.

Unique molecular pathways contribute to radiation response in male and female primary human GBM cell lines

The authors performed irradiation dose response curves with 4 male and 5 female primary human GBM lines and found that there was no significant difference in the median IC50 values for male and female lines (Fig. 1a). Based on previous analysis, they identified a set of differentially regulated genes associated with better survival in male cluster 5 (MC5) or female cluster 3 (FC3) GBM patients. Pathway analysis of these gene sets identified enrichment of a cell cycle regulation pathway in males (MC5 – 17 genes) and an integrin signaling pathway in females (FC3 – 9 genes) (Suppl. Table 2). This presents a major difference between the same human cancer cell types with no common genes/pathways. Can the authors comment on this?

Female mouse GBM model astrocytes are more sensitive to radiation treatment

In order to better understand the mechanisms underlying radiation response in male and female GBM, the authors utilized an in vitro mouse model consisting of murine astrocytes with loss of function of the tumor suppressors Nf1 and p53 (Nf1^{-/-} DNp53). When implanted intracranially, both male and female cells form tumors that histologically resemble high grade gliomas, although the frequency of tumor formation differs by cell sex. Additionally, this model displays sex differences in gene expression that are concordant with sex differences in human GBM patient gene expression. Irradiation dose response curves with male and female Nf1^{-/-} DNp53 astrocytes, revealed a sex difference, with female cells being more sensitive to irradiation (Fig. 2), consistent with results from human GBM that suggest female GBM patients are more responsive to standard of care therapy (radiation and chemotherapy). The authors also showed irradiation-induced DNA damage as documented by gammaH2AX foci with a decline in both male and female cells, which was more rapid in male cells (Figs 2 e & d), suggesting that radiation treatment results in more persistent DNA damage in female cells

Is this due to more efficient repair in male vs female cells taking into account that both cells are mutant p53, and is this linked to differentially expressed genes (ST2)? Can the authors comment on this?

Then, the authors investigated the expression of cleaved caspase-3 and cleaved PARP as a measure of apoptosis in both male and female cells following irradiation and suggested that apoptosis does not appear to be the primary mechanism responsible for the decrease in cell number following irradiation (Fig. 3 & SF2). However, to further support such a conclusion, a flow cytometric analysis should be performed using annexin V/PI staining.

Next, the authors evaluated irradiation-induced senescence in both male and female cells, 5 days after irradiation, and showed that there was a clear, dose dependent increase in the percentage of senescent cells in both male and female Nf1^{-/-} DNp53 mouse astrocytes (Fig. 3).

There is some discrepancy in the conditions used. For example, for survival, doses of 3 & 9 Gy are used and also 3 Gy for the induction of DNA damage (Fig. 2) and higher doses (6 & 8 Gy) for apoptosis and senescence (Fig. 3). While both male and female cells recover within 48 hrs from irradiation-induced DNA damage with 3 Gy, albeit female cells more slowly (Fig. 2), senescence is evaluated following treatment with 6 and 8 Gy and after 5 days, and the number of blue (SA-beta-Gal) positive cells are very low. I am not convinced, and the authors should perform the experiments with the same doses of irradiation, and also repeat the SA-beta-gal staining. The same for Ki67. It should be repeated with 3 Gy as the experiments done for gammaH2AX. Alternatively, they should do all gammaH2AX, SA-beta-gal and Ki67 at 3, 6, and 8 or 9 Gy, so that the results will be comparable. In my opinion 3Gy will be the best dosage since it induces DNA damage most likely without apoptotic events (which they should show it by flow cytometry Annexin V/PI staining).

Flow cytometry can also be performed with PI to show cell cycle distribution in a dose and time-dependent manner and can also better explain their finding on the expression of CcnD1 and p27 (Fig. 3F).

The authors concluded that senescence is a central component of the response to irradiation in Nf1^{-/-} DNp53 astrocytes, and that this, rather than apoptosis, primarily drives the decrease in cell growth after radiation.

Expression of p21, 24 hours after irradiation correlates with the senescence response observed at 5 days

Next, the authors investigated the expression of the CDKI p21 and its link to irradiation-induced senescence (Fig. 4). The authors state that: While the Nf1^{-/-} DNp53 astrocytes express a dominant negative p53, and thus lack p53 function, they retain the ability to upregulate p21 in response to DNA damage, presumably through p53-independent mechanisms. First, several p53 dominant negative mutants retain their ability to induce p53-target genes and other effects of the wtp53, hence cells expressing DNp53 have a functional mutant p53 and they are not like p53-null cells, so their cells express a DNp53 protein and both p53-dependent and -independent mechanisms may be involved in the induction of p21. So they should investigate the expression of p53 in response to irradiation alongside with p21 in a dose dependent manner (0, 3, 6, 8/9), which should be investigated both at mRNA (qPCR) and protein levels. If they wish to state, p53-independent mechanisms, then they should knock-down p5 expression. To also clarify this, for example, they can use the NF1/53 CRISPR astrocytes (see SF6) to investigate p21 expression.

Expression of p21 differentially correlates with SA-beta-gal positivity in male and female mouse GBM model astrocytes

The authors suggested that p21/Cdk2 ratio plays a role in the maintenance of senescence following irradiation, and that there may be a sex difference in sensitivity to p21/Cdk2 levels, and/or that p21 may be playing a greater role in irradiation induced senescence in female GBM cells, than in male GBM cells. This conclusion comes from an observation based on the ratio. To conclusively state this, the authors need to knock-down CDk2.

Wild-type mouse astrocytes exhibit sex differences in the relationship between p21 and SA-beta-gal. Untreated normal male and female astrocytes exhibit high levels of SA-beta-Gal staining, suggesting a senescent response (Fig. 5a). Can the authors comment on this?

This section concludes that normal and GBMs irrespective of their cancer-related mutations undergo p21-dependent senescence in response to irradiation.

Sex differences in senescence are observed in wild-type astrocytes with repeated in vitro passaging
The authors use terms like low passage (2 or p3) and high passage (p5). Passage is not a term that should be used in senescence experiments as it is very confusing and it should be really related to population doublings (PDLs). For example, the difference between p2/p3 and p5 is very small, but if one passage refers to a 1:2 split ratio this adds 1 PDL but if the cells are culture at 1:16 split ratio then this adds 4 PDLs. So, what is really the lifespan of male and female mouse astrocytes in terms of PDLs? This is important as the authors claim that female wild-type astrocytes have increased senescent cells at later passages (SF4) which is really not obvious from SA-beta-Gal staining. Can this difference also be an artifact of cell culture conditions?

I wouldn't expect to see any differences by flow cytometry between young vs senescence cells in culture except perhaps a slight increase in G1 which in most cases, varies. However, growth curves may be better to reveal differences in the growth rates of male and female cells.

Primary human GBM lines exhibit sex differences in the relationship between p21 and SA-beta-gal
The primary human GBM cell lines presented in supplementary table 1 have different genotypes including oncogenic mutations (e.g. EGFR) and inactivating TS mutations (e.g. p53). I assume that in Fig. 6a, the authors present collective data which is not appropriate, rather they should present data for SA-beta-Gal and gammaH2AX in at least 2 mutant cell lines (e.g. B49 and B66) and 2 'wild-type'

(e.g. B178, B30), and the data presented in Fig. 6b, both at RNA and protein levels.

Knockdown of p21 decreases senescence in irradiated female mouse GBM model astrocytes
Using CRISPR/Cas9 technology, the authors knocked-down p21 in transgenic mouse astrocytes and showed that irradiation-induced senescence was p21-dependent (Fig. 7). Does this affect the formation of gammaH2AX foci in response to irradiation? These experiments should be extended to human GBM cell lines.

In the last results section, the authors investigated gonadal sex patterns the relationship between p21 and SA-beta-gal positivity using an FCG GBM model. Naturally, I don't understand why the used CRISPR/Cas9 to delete Nf1 and p53 from these astrocytes, to mimick their Nf1-/- DNp53 GBM model. First, DNp53 astrocytes express should express p53 as they bear a DN p53, whereas the mouse model is p53-null. So these are two different conditions, and in both cases p21 is induced which means that senescence is p53-independent. Secondly, normal mouse astrocytes have the same behaviour to Nf1-/- DNp53 in response to radiation. These are confusing experiments. Probably, it would have been better to use CRISPR/Cas9 to ablate p21.

While male GBMs are more frequent than female GBMs, and that therapy is more effective in the latter, statistically there is no difference in overall survival. I am not convinced that the differences in the responses to radiation of male vs female GBMs and astrocytes is due to p21-mediated senescence (see also ST2). For example, why females are more sensitive to subtle changes in p21 levels? Can genes involved in integrin signalling account for the responses of female GBMs to irradiation rather than p21? I think, by employing treated and radiation-treated patient-derived human GBM cell lines such as those I suggested above, and deep transcriptomics analysis (or proteomics in case for posttranslational modifications), it may be possible to identify critical genes and pathways involved in the responses of male and female GBM cells to irradiation.

Reviewer #1 (Remarks to the Author):

The methods they apply are cutting-edge and their conclusions sound. The manuscript is elegant, fluent and highly pleasant-to-read.

We thank the reviewer for their positive comments.

1. The start comes somewhat abrupt. I understand that the authors build on gene signatures and clusters, which they have previously identified in human m/f tumor samples and confirmed in chemotherapy-treated m/f cells. Still, it is a very supervised approach with moderate, not too intuitive results that do not fully connect to previous findings. Are there independent radiotherapy-induced m/f-specific gene signatures?

This is a very helpful comment and exciting suggestion. We agree that a signature of radio-response in male and female patients would be highly valuable. Unfortunately, we have no matched patient specimens from prior to their irradiation and either during or immediately post treatment, and only a few matched pre- and long-term post-specimens. In addition, the analysis is likely to require many specimens to be adequately powered for the detection of significant sex-biased responses. We could pursue this work in murine models but would want to acquire multiple timepoints to rigorously examine the time-course and dynamic range of male/female gene expression responses. For these reasons, we thank the reviewer for the idea, which we hope to pursue, but this will constitute a project of its own that is beyond the scope of the current study.

We have amended the text to reduce the abruptness in the start of the results section. Specifically, we now state the following (Results, p4):

“Female GBM patients exhibit significantly longer median survival than male patients^{5,15}; concordantly, we previously determined that, compared to males, female GBM patients exhibited greater initial therapeutic response to radiation and temozolomide therapy⁷. Moreover, the degree of initial response to chemoradiation was significantly correlated with survival in female, but not male, patients. To investigate the molecular features underlying this sex difference in treatment response, we applied a Joint and Individual Variance Explained (JIVE) algorithm to identify and filter out gene expression patterns that were shared between male and female patients. This revealed unique male and female gene signatures that correlated with sex differences in survival, as well as with sensitivity to a range of chemotherapeutic agents in primary human GBM cell lines. Since our long-term goal is to determine whether survival with GBM can be enhanced by adapting treatment to sex differences in biology, to further explore the clinical relevance of these gene signatures, we investigated whether they similarly correlated with sensitivity to radiation.”

2. The authors emphasize the relevance of using correlations as a statistical means to identify relevant biological associations. However, when looking at several scatter plots, e.g. in Figure 6, some correlations are driven by outliers and the exclusion of single data points would result in different results. I do believe that their findings are true since the same associations are found in non-neoplastic and neoplastic murine and human cells but it should be addressed as a potential limitation.

This is a critical point. Due to the small sample sizes, the observed correlations do rely on a few points on the extreme of the x-axis. These points, however, are not considered outliers, but rather influential

leverage points in statistical terminology. Our reasoning is as follows: an outlier point usually doesn't follow the y versus x overall (linear) relationship. The leverage points however, usually do and are characterized by having "extreme" x and y values. Here, expression levels and the extreme values were caused by the fact that cells have varying sensitivity to treatment resulting in the dynamic range of responses and x values, and in turn subsequently leading to the extreme of y values on the far ends of the linear line. In addition, as the reviewer said, we observed these relationships consistently, across multiple models, indicating this is not due to outliers, but reflects accurate observations. We have amended the text to communicate this point (Results, p15):

"Given the small number of human GBM specimens, we considered whether these correlations were being driven by outlier versus leverage points²⁸. We concluded that these were leverage points as they were characterized by having "extreme" x and y values that followed the overall linear relationship. In addition, they were consistent with what we observed in the wild type and transformed murine models."

Reviewer #2 (Remarks to the Author):

This is an excellent study from an outstanding group who are leaders in the sex differentiation stratification of GBM patient outcomes. The experiments are well-controlled and the data are interpreted wisely.

We thank the reviewer for their positive comments.

There is only one comment to address here. Does any of this matter? It's already known that female patients get GBM less frequently than men, but in the current study, the authors primarily evaluate senescence parameters in response to irradiation (IR). In vivo analysis of whether there are differential growth patterns after IR with female vs. male GBM may allay this concern - or it could corroborate it. Either way, in vivo analysis of the currently in vitro study needs to be performed to support the in vitro outcomes that are in the current report.

This is the question and a primary motivation for our work. Sex differences in normal and cancer biology are easy to detect. Determining which sex differences in biology matter clinically to incidence, response to treatment, and survival, is what we hope our ongoing studies will help to determine. In addition to known sex differences in GBM incidence, there are sex differences in progression-free survival after standard-of-care treatment. We directly demonstrated that female patients with GBM respond better to radiation and temozolomide treatment as compared to male patients (Yang, *Science Translational Medicine* (2019)). It is the clinical difference that motivated this study to explore contributing mechanisms. We believe the data matter to these clinical questions, and agree that *in vivo* data will be critical in this line of investigation.

We undertook an examination of radiation response in intracranial xenografts of male and female murine GBM

model astrocytes. We have not previously evaluated the effects of radiation in this model. Thus, we made several assumptions in an attempt to complete a study that might contribute to this revision. We planned intracranial implantation of *Nf1*^{-/-} *DNp53* murine astrocytes into 40 male and 40 female mice. Male mice received male cell implants and female mice received female cell implants. Using weekly bioluminescence imaging, we first identified male and female mice with steadily growing tumors until they reached a threshold of 5-fold increase compared to the first week post-implantation. 36/40 male and 19/40 female mice met this criterion. These animals were then randomized to treatment with 2 Gy/day for five days irradiation. Animals were euthanized when they had lost more than 10% of their initial body mass, developed neurological symptoms, or appeared ill. The date of euthanasia was used in the survival analysis. As you can see in the accompanying figure for *review purposes only*, there is a strong trend towards significant response to irradiation in the female, but not in the male mice. Based on these initial data we performed a power calculation to determine how many animals we would need to detect a significant difference of this magnitude and concluded we would need to observe 180 events (deaths). We feel an experiment of this size is beyond the scope of the current study.

To better communicate the importance of this study we have amended the first paragraph of the results (Results, p4):

“Female GBM patients exhibit significantly longer median survival than male patients^{5,15}; concordantly, we previously determined that, compared to males, female GBM patients exhibited greater initial therapeutic response to radiation and temozolomide therapy⁷. Moreover, the degree of initial response to chemoradiation was significantly correlated with survival in female, but not male, patients. To investigate the molecular features underlying this sex difference in treatment response, we applied a Joint and Individual Variance Explained (JIVE) algorithm to identify and filter out gene expression patterns that were shared between male and female patients. This revealed unique male and female gene signatures that correlated with sex differences in survival, as well as with sensitivity to a range of chemotherapeutic agents in primary human GBM cell lines. Since our long-term goal is to determine whether survival with GBM can be enhanced by adapting treatment to sex differences in biology, to further explore the clinical relevance of these gene signatures, we investigated whether they similarly correlated with sensitivity to radiation.”

Reviewer #3 (Remarks to the Author):

Unique molecular pathways contribute to radiation response in male and female primary human GBM cell lines

The authors performed irradiation dose response curves with 4 male and 5 female primary human GBM lines and found that there was no significant difference in the median IC50 values for male and female lines (Fig. 1a). Based on previous analysis, they identified a set of differentially regulated genes associated with better survival in male cluster 5 (MC5) or female cluster 3 (FC3) GBM patients. Pathway analysis of these gene sets identified enrichment of a cell cycle regulation pathway in males (MC5 – 17 genes) and an integrin signaling pathway in females (FC3 – 9 genes) (Suppl. Table 2). This presents a major difference between the same human cancer cell types with no common genes/pathways. Can the authors comment on this?

In the prior analysis we identified a large component of gene expression that was shared between male and female glioblastoma patients overall and a calcium signaling pathway that was shared between the longest surviving male and female patients. The analysis we performed in that prior study, Joint and

Individual Variance Explained (JIVE) was specifically designed to remove the shared component so that we could have greater sensitivity in an analysis of what distinguished male and female glioblastoma patients. The gene signatures we utilized in Figure 1 distinguished the longest surviving females from all other females or longest surviving males from all other males. We have amended the text to better clarify this important point (Results, p4):

“Female GBM patients exhibit significantly longer median survival than male patients^{5,15}; concordantly, we previously determined that, compared to males, female GBM patients exhibited greater initial therapeutic response to radiation and temozolomide therapy⁷. Moreover, the degree of initial response to chemoradiation was significantly correlated with survival in female, but not male, patients. To investigate the molecular features underlying this sex difference in treatment response, we applied a Joint and Individual Variance Explained (JIVE) algorithm to identify and filter out gene expression patterns that were shared between male and female patients. This revealed unique male and female gene signatures that correlated with sex differences in survival, as well as with sensitivity to a range of chemotherapeutic agents in primary human GBM cell lines. Since our long-term goal is to determine whether survival with GBM can be enhanced by adapting treatment to sex differences in biology, to further explore the clinical relevance of these gene signatures, we investigated whether they similarly correlated with sensitivity to radiation.”

Female mouse GBM model astrocytes are more sensitive to radiation treatment

In order to better understand the mechanisms underlying radiation response in male and female GBM, the authors utilized an in vitro mouse model consisting of murine astrocytes with loss of function of the tumor suppressors Nf1 and p53 (Nf1^{-/-} DNp53). When implanted intracranially, both male and female cells form tumors that histologically resemble high grade gliomas, although the frequency of tumor formation differs by cell sex. Additionally, this model displays sex differences in gene expression that are concordant with sex differences in human GBM patient gene expression. Irradiation dose response curves with male and female Nf1^{-/-} DNp53 astrocytes, revealed a sex difference, with female cells being more sensitive to irradiation (Fig. 2), consistent with results from human GBM that suggest female GBM patients are more responsive to standard of care therapy (radiation and chemotherapy). The authors also showed irradiation-induced DNA damage as documented by gammaH2AX foci with a decline in both male and female cells, which was more rapid in male cells (Figs 2 e & d), suggesting that radiation treatment results in more persistent DNA damage in female cells. Is this due to more efficient repair in male vs female cells taking into account that both cells are mutant p53, and is this linked to differentially expressed genes (ST2)? Can the authors comment on this?

This is an important and incompletely understood point. Considering this comment has highlighted a better approach to reporting these data, which we have used to amend the text. We do not yet know how the gene signatures relate to the activation and maintenance of DNA damage response activation, nor do we know whether the degree of repair to turn off the response is the same in male and female cells. What we do know is that γ H2AX foci can persist after resolution of double strand breaks. Thus, the γ H2AX staining is most valuable as a measure of initial DNA damage and the duration of DDR activation, rather than as a measure of the efficiency or completeness of repair. As persistent DDR activation promotes cell cycle arrest and senescence, this could well relate to sex differences in the senescence response to irradiation. We have made the following addition to the text to address this important point (Results, p6-7).

“Finally, we assessed activation of the DNA damage response (DDR) pathway by irradiating male and female Nf1^{-/-} DNp53 astrocytes with 3 or 8 Gy and then performing

immunofluorescence for γ H2AX, a marker of cellular response to DNA double strand breaks (Fig. 2d, e). We quantified the percent of cells with greater than 10 γ H2AX foci at 1, 6, 24, and 48 hours after irradiation (Fig. 2e). At 1 hour after treatment, essentially 100% of male and female cells were positive. Over time, the percent of positive cells declined significantly more rapidly and more completely in male compared to female cells. At 48 hours, approximately 10% of male cells treated with 3 or 8 Gy irradiation had greater than 10 γ H2AX foci. In contrast, female cells treated with 3 Gy or 8 Gy irradiation still had 25% and 15% positivity for γ H2AX foci, respectively, at 48 hrs. Moreover, while radiation dose had no significant effect on the male cell response, there was a significant difference in female cell response to 3 versus 8 Gy. This suggests that radiation treatment results in more persistent DDR activation in female cells. As γ H2AX foci can persist after resolution of double-strand breaks¹⁶, differences in γ H2AX staining do not necessarily reflect differences in the efficiency or completeness of DNA repair. The results do, however, indicate that there are sex differences in DDR pathway activation and suggest that persistent DDR activation may contribute to enhanced radiation sensitivity in female cells.”

Then, the authors investigated the expression of cleaved caspase-3 and cleaved PARP as a measure of apoptosis in both male and female cells following irradiation and suggested that apoptosis does not appear to be the primary mechanism responsible for the decrease in cell number following irradiation (Fig. 3 & SF2). However, to further support such a conclusion, a flow cytometric analysis should be performed using annexin V/PI staining.

As suggested, we performed annexin quantification by flow cytometry and the results are presented in a revised Figure 3. Whereas at 5 days after irradiation, male cells exhibited a significantly greater increase in caspase-3 cleavage, Annexin V staining indicated that there was significantly greater positivity in female cells compared to male cells. As Annexin V staining is now recognized to be a feature of both apoptotic and non-apoptotic cell death, we conclude that non-apoptotic cell death could contribute to sex differences in radiation response. These new results are presented in revised Figure 3b and revised Supplemental Figure 2. The changes to the Results section text (p7-8) are presented below. There are accompanying text changes to the Figure Legend (p39) and Materials and Methods (p30) as well as to the Supplemental Material.

“Apoptosis is only one potential mechanism by which cells can die following irradiation. In order to look more quantitatively at cell death, we measured annexin V staining by flow cytometry 24 hours and five days after irradiation with 0, 3, 6, or 8 Gy (Fig. 3b, Supplementary Fig. 2c, d). We observed a dose-dependent increase in the annexin V positive fraction at both 24 hours (Supplementary Fig. 2d) and 5 days (Fig. 3b) after irradiation. At 24 hours there was no significant effect of sex on annexin V staining, while at 5 days females had higher levels of annexin V positivity. As cell surface exposure of phosphatidylserine is now recognized to be a feature of both apoptotic and non-apoptotic modes of cell death¹⁸, the annexin V staining, together with the cleaved caspase-3 results, suggests that irradiation leads to increased non-apoptotic cell death in females compared to males, which could contribute to greater female radiation sensitivity.”

There is some discrepancy in the conditions used. For example, for survival, doses of 3 & 9 Gy are used and also 3 Gy for the induction of DNA damage (Fig. 2) and higher doses (6 & 8 Gy) for apoptosis and senescence (Fig. 3). While both male and female cells recover within 48 hrs from irradiation-induced DNA damage with 3 Gy, albeit female cells more slowly (Fig. 2), senescence is evaluated following treatment with 6 and 8 Gy and after 5 days, and the number of blue (SA-beta-Gal) positive cells are very low. I am not convinced, and the authors should perform the experiments with the same doses of irradiation, and also repeat the SA-beta-gal staining. The same for Ki67. It should be repeated with 3 Gy as the experiments done for gammaH2AX.

We have now redone all key experiments with 0, 3, 6, and 8 Gy treatments and present these new data in 27 new panels of data. These changes to the manuscript are described in detail below.

Alternatively, they should do all γ H2AX, SA-beta-gal and Ki67 at 3, 6, and 8 or 9 Gy, so that the results will be comparable. In my opinion 3Gy will be the best dosage since it induces DNA damage most likely without apoptotic events (which they should show it by flow cytometry Annexin V/PI staining). Flow cytometry can also be performed with PI to show cell cycle distribution in a dose and time-dependent manner and can also better explain their finding on the expression of CcnD1 and p27 (Fig. 3F).

In order for the data to be comparable, we have redone the γ H2AX (revised Figure 2e), SA-beta-gal (revised Figure 3c) and Ki67 (revised Figure 3d and revised Supplemental Figure 4a) at 0, 3, 6, and 8 Gy. We have also performed Annexin V and Edu incorporation by flow cytometry as a function of radiation dose. The Annexin results were described in response to the prior comment. The new Edu data are presented in revised Figure 3e and revised Supplemental Figure 4. EdU incorporation significantly declines in both male and female cells as a function of radiation dose. The decline is significantly more rapid and to a greater extent in female cells concordant with the Ki67 results (revised Figure 3d, which now includes 3 Gy) and the growth curves (Figure 2b). The change to the Results section describing the EdU results (p8) are below. There are accompanying text changes to the Figure Legend (p40) and Materials and Methods (p29-30).

“As senescence in response to DNA damage involves cell cycle arrest, we directly measured the effects of irradiation on DNA synthesis using flow cytometry to measure EdU incorporation. Concordant with the Ki67 and regrowth after irradiation (Fig. 2b) data, female cells exhibited a continuous decline in EdU incorporation over the entire dose range, while male cells were relatively resistant to 3 Gy and only exhibited substantial arrest at 8 Gy (Fig. 3e, Supplementary Fig. 4b).”

First, several p53 dominant negative mutants retain their ability to induce p53-target genes and other effects of the wtp53, hence cells expressing DNp53 have a functional mutant p53 and they are not like p53-null cells, so their cells express a DNp53 protein and both p53-dependent and –independent mechanisms may be involved in the induction of p21. So they should investigate the expression of p53 in response to irradiation alongside with p21 in a dose dependent manner (0, 3, 6, 8/9), which should be investigated both at mRNA (qPCR) and protein levels.

To directly address this point, we performed western blot analysis for endogenous wildtype p53 expression 24 hours and 5 days after irradiation with 0,3,6, or 8 Gy. We found that while p21 expression exhibited radiation dose-dependent increases, no such increase was observed for the endogenous p53. These data are presented in a new Supplementary Figure 5. These results are consistent with our previously published results (Sun *JCI* 2014). We have amended the results section (p10) as follows.

“To confirm that increased p21 expression was independent of the wildtype p53 alleles in our cell model, we directly measured the endogenous p53 and p21 protein levels 24 hours and five days following irradiation with 0, 3, 6, or 8 Gy. Whereas p21 expression was induced by irradiation in male and female Nf1-/- DNp53 cells, there was no concomitant induction of p53 (Supplementary Fig. 5).”

If they wish to state, p53-independent mechanisms, then they should knock-down p53 expression. To also clarify this, for example, they can use the NF1/53 CRISPR astrocytes (see SF6) to investigate p21

expression.

We disrupted p53 function in our murine models using the dominant negative construct as well as CRISPR editing and observed the same effect. These results were similar to the results obtained with patient-derived GBM cell lines, in which p53 function was disrupted by multiple alterations. Thus, as the reviewer observed in a later comment (*So these are two different conditions, and in both cases p21 is induced which means that senescence is p53-independent.*), we see a similar relation between p21 expression and senescence regardless of how we disrupt p53 or in which models. Therefore, this mechanism is p53 independent. We have amended the Discussion to better communicate the basis for this conclusion (p19)

“Importantly, the relationship between SA-beta-gal and p21 was evident across the heterogeneous p53 status of the varying murine models (wildtype, dominant negative, CRISPR deletion) and the heterogeneous nature of the human GBM cell lines. Together, these results indicate that the sex differences in p21-induced senescence are p53 independent.”

Expression of p21 differentially correlates with SA-beta-gal positivity in male and female mouse GBM model astrocytes. The authors suggested that p21/Cdk2 ratio plays a role in the maintenance of senescence following irradiation, and difference in sensitivity to that p21 may be playing a role in induced senescence in male GBM cells. This observation based on the this, the authors need to

that there may be a sex p21/Cdk2 levels, and/or greater role in irradiation female GBM cells, than in conclusion comes from an ratio. To conclusively state knock-down CDK2.

We attempted CDK2 knockdown in our murine lines. As can be seen in the second figure for review purposes only, CDK2 knockdown resulted in an unexpected and consistent inhibition of p21 expression, suggesting a homeostatic response. Thus, the effects of CDK2 knockdown require additional studies that we feel are beyond the scope of this current manuscript.

Wild-type mouse astrocytes exhibit sex differences in the relationship between p21 and SA-beta-gal. Untreated normal male and female astrocytes exhibit high levels of SA-beta-Gal staining, suggesting a senescent response (Fig. 5a). Can the authors comment on this?

Wildtype astrocytes have limited proliferative potential in culture and the high rate of senescence is reflective of that. We have added comments to the results section (p 12) to clarify this point:

“Even in the untreated condition, wildtype astrocytes exhibited measurable levels of senescence, reflective of their limited replicative potential (Fig. 5a).”

This section concludes that normal and GBMs irrespective of their cancer-related mutations undergo p21-dependent senescence in response to irradiation.

Sex differences in senescence are observed in wild-type astrocytes with repeated in vitro passaging. The authors use terms like low passage (2 or p3) and high passage (p5). Passage is not a term that should be used in senescence experiments as it is very confusing and it should be really related to population doublings (PDLs). For example, the difference between p2/p3 and p5 is very small, but if one passage refers to a 1:2 split ratio this adds 1 PDL but if the cells are culture at 1:16 split ratio then this adds 4 PDLs. So, what is really the lifespan of male and female mouse astrocytes in terms of PDLs? This is important as the authors claim that female wild-type astrocytes have increased senescent cells at later passages (SF4) which is really not obvious from SA-beta-Gal staining. Can this difference also be an artifact of cell culture conditions? I wouldn't expect to see any differences by flow cytometry between young vs senescence cells in culture except perhaps a slight increase in G1 which in most cases, varies. However, growth curves may be better to reveal differences in the growth rates of male and female cells.

These are excellent points. We have corrected the figure to put it in terms of population doubling levels. We routinely split 1:5. Male and female wildtype astrocytes proliferate at the same rate and thus each passage is approximately 2.25 doublings for each. The Figure is now presented in terms of population doublings and the text has been amended to report the findings in terms of population doublings.

Primary human GBM lines exhibit sex differences in the relationship between p21 and SA-beta-gal. The primary human GBM cell lines presented in supplementary table 1 have different genotypes including oncogenic mutations (e.g. EGFR) and inactivating TS mutations (e.g. p53). I assume that in Fig. 6a, the authors present collective data which is not appropriate, rather they should present data for SA-beta-Gal and gammaH2AX in at least 2 mutant cell lines (e.g. B49 and B66) and 2 'wild-type' (e.g. B178, B30), and the data presented in Fig. 6b, both at RNA and protein levels.

We respectively disagree with the appropriateness of a presentation of two wildtype and p53 mutant male and female lines in this analysis. First, TP53 function is altered in GBM through multiple mechanisms including alteration of the *TP53* gene, amplification of *MDM2*, or loss of *ARF*. Moreover, missense mutations of the TP53 DNA binding domain result in a loss of canonical transcriptional activity and variable gain of aberrant oncogenic function. Therefore, whether the p53 gene is mutated or not does not directly correlate with TP53 function. Second, as mentioned by the reviewer, genetic heterogeneity in GBM is well-described and broadly affects cellular functions including cell cycle regulation, DNA repair, and cell death pathways, each of which are directly involved in radiation response. Therefore, a display item that separated four male and four female patient-derived cell lines into two p53 wildtype and mutants for comparison would not be conclusive with regard to the relationship between p53 status, p21, sex, and radiation-induced senescence. The purpose of these patient-derived cell line studies was to determine whether the human lines exhibited a similar effect of sex on the relationship between p21 and SA-beta-gal as was seen in the murine models.

Knockdown of p21 decreases senescence in irradiated female mouse GBM model astrocytes. Using CRISPR/Cas9 technology, the authors knocked-down p21 in transgenic mouse astrocytes and showed that irradiation-induced senescence was p21-dependent (Fig. 7). Does this affect the formation of gammaH2AX foci in response to irradiation? These experiments should be extended to human GBM cell lines.

We evaluated the effects of p21 knockdown on gammaH2AX foci at 0, 3 and 8 Gy. We found that p21 knockdown had no effect on gammaH2AX foci resolution in male cells but significantly altered the

resolution in females rendering it nearly identical to that in males. These new data are presented in revised Figure 7e-g. The Results text has been amended as follows (p15-16)

“To determine whether p21 knockdown would also exhibit a sex-biased effect on DDR, we quantified γ H2AX foci resolution in Cas9 control and p21 KD Nf1-/- DNp53 astrocytes following irradiation with either 3 or 8 Gy. Consistent with our findings in Fig. 2e, female Cas9 control cells exhibited sustained γ H2AX foci compared to male Cas9 control cells (Fig. 7e). Similar to the effects of p21 knockdown on senescence, the kinetics and magnitude of γ H2AX foci resolution following irradiation with 3 or 8 Gy was unaffected by p21 status in male cells (Fig. 7f). In contrast, the kinetics and magnitude of γ H2AX foci resolution after treatment with 8 Gy were significantly increased in female cells by p21 KD (Fig. 7g) to male comparable levels. There was a trend towards an effect of p21 KD in female cells irradiated with 3 Gy, however this did not reach statistical significance. Together, these data support the idea that p21 expression contributes significantly to the levels of senescence after irradiation in females, but not in males. Notably, despite the decrease in SA- β -gal in female p21 KD cells, they still retain a significant senescence response, and in fact decrease only to the levels of senescence in male cells. This suggests that additional pathways, beyond p21, are contributing to senescence after irradiation in both male and female cells.”

We also attempted to knockdown p21 in the human lines multiple times but were unable to propagate any of the lines following p21 knockdown. We concluded that achieving p21 knockdown in these GBM lines will require work beyond the scope of this revision.

In the last results section, the authors investigated gonadal sex patterns the relationship between p21 and SA-beta-gal positivity using an FCG GBM model. Naturally, I don't understand why the used CRISPR/Cas9 to delete Nf1 and p53 from these astrocytes, to mimick their Nf1-/- DNp53 GBM model. First, DNp53 astrocytes express should express p53 as they bear a DN p53, whereas the mouse model is p53-null. So these are two different conditions, and in both cases p21 is induced which means that senescence is p53-independent. Secondly, normal mouse astrocytes have the same behaviour to Nf1-/- DNp53 in response to radiation. These are confusing experiments. Probably, it would have been better to use CRISPR/Cas9 to ablate p21.

While male GBMs are more frequent than female GBMs, and that therapy is more effective in the latter, statistically there is no difference in overall survival. I am not convinced that the differences in the responses to radiation of male vs female GBMs and astrocytes is due to p21-mediated senescence (see also ST2).

We agree that there are no statistically significant differences in overall survival in males and females with GBM. There are however statistically significant differences in median survival. It is reasonable to interpret this as differences in response to upfront radiation and temozolomide therapy. We directly demonstrated this to be true in our STM paper. We reported that sex-biased gene expression signatures were correlated with chemotherapy IC₅₀ and here we are extending this to radiation therapy response as this is the cornerstone of treatment. We have amended the Introduction to better reflect the rationale for these studies (p4).

“Female GBM patients exhibit significantly longer median survival than male patients^{5,15}; concordantly, we previously determined that, compared to males, female GBM patients exhibited greater initial therapeutic response to radiation and temozolomide therapy⁷. Moreover, the degree of initial response to chemoradiation was significantly correlated with survival in female, but not male, patients. To investigate the molecular features underlying this sex difference in treatment response, we applied a

Joint and Individual Variance Explained (JIVE) algorithm to identify and filter out gene expression patterns that were shared between male and female patients. This revealed unique male and female gene signatures that correlated with sex differences in survival, as well as with sensitivity to a range of chemotherapeutic agents in primary human GBM cell lines. Since our long-term goal is to determine whether survival with GBM can be enhanced by adapting treatment to sex differences in biology, to further explore the clinical relevance of these gene signatures, we investigated whether they similarly correlated with sensitivity to radiation.”

For example, why females are more sensitive to subtle changes in p21 levels? Can genes involved in integrin signalling account for the responses of female GBMs to irradiation rather than p21? I think, by employing treated and radiation-treated patient-derived human GBM cell lines such as those I suggested above, and deep transcriptomics analysis (or proteomics in case for posttranslational modifications), it may be possible to identify critical genes and pathways involved in the responses of male and female GBM cells to irradiation.

We agree completely and the suggested studies will be the next stage of this work.

REVIEWERS' COMMENTS:

Reviewer #1 (Remarks to the Author):

In their revised version of the manuscript the authors have adequately and concisely addressed the points I raised during initial review. Overall, they made significant efforts to improve the scientific quality and impact of the study.

I have no further remarks.

Reviewer #3 (Remarks to the Author):

I am satisfied with the authors replies to my comments and questions, and the fact that the authors did additional experiments in support of their paper results, discussion and conclusions and to improve its quality.